# Identification of c-di-GMP/FleQ-Regulated New Target Genes, Including *cyaA*, Encoding Adenylate Cyclase, in *Pseudomonas putida*

Yujie Xiao,[a] Haozhe Chen,[a] Liang Nie,[a] Meina He,[a] Qi Peng,[a] Wenjing Zhu,[a] Hailing Nie,[a] (ID) Wenli Chen,[a] (ID) Qiaoyun Huang[a,b]

[a]State Key Laboratory of Agricultural Microbiology, Huazhong Agricultural University, Wuhan, China
[b]Hubei Key Laboratory of Soil Environment and Pollution Remediation, College of Resources and Environment, Huazhong Agricultural University, Wuhan, China

**ABSTRACT** The bacterial second messenger cyclic diguanylate (c-di-GMP) modulates plankton-to-biofilm lifestyle transition of *Pseudomonas* species through its transcriptional regulatory effector FleQ. FleQ regulates transcription of biofilm- and flagellum-related genes in response to c-di-GMP. Through transcriptomic analysis and FleQ-DNA binding assay, this study identified five new target genes of c-di-GMP/FleQ in *P. putida*, including *PP_0681*, *PP_0788*, *PP_4519* (*lapE*), *PP_5222* (*cyaA*), and *PP_5586*. Except *lapE* encoding an outer membrane pore protein and *cyaA* encoding an adenylate cyclase, the functions of the other three genes encoding hypothetical proteins remain unknown. FleQ and c-di-GMP coordinately inhibit transcription of *PP_0788* and *cyaA* and promote transcription of *PP_0681*, *lapE*, and *PP_5586*. Both *in vitro* and *in vivo* assays show that FleQ binds directly to promoters of the five genes. Further analyses confirm that LapE plays a central role of in the secretion of adhesin LapA and that c-di-GMP/FleQ increases *lapE* transcription, thereby promoting adhesin secretion and biofilm formation. The adenylate cyclase CyaA is responsible for synthesis of another second messenger, cyclic AMP (cAMP). FleQ and c-di-GMP coordinate to decrease the content of cAMP, suggesting that c-di-GMP and FleQ coregulate cAMP by modulating *cyaA* expression. Overall, this study adds five new members to the c-di-GMP/FleQ-regulated gene family and reveals the role of c-di-GMP/FleQ in LapA secretion and cAMP synthesis regulation in *P. putida*.

**IMPORTANCE** c-di-GMP/FleQ promotes the plankton-to-biofilm lifestyle transition at the transcriptional level via FleQ in *Pseudomonas* species. Identification of new target genes directly regulated by c-di-GMP/FleQ helps to broaden the knowledge of c-di-GMP/FleQ-mediated transcriptional regulation. Regulation of *lapE* by c-di-GMP/FleQ guarantees highly efficient LapA secretion and biofilm formation. The mechanism of negative correlation between c-di-GMP and cAMP in both *P. putida* and *P. aeruginosa* remains unknown. Our result concerning transcriptional inhibition of *cyaA* by c-di-GMP/FleQ reveals the mechanism underlying the decrease of cAMP content by c-di-GMP in *P. putida*.

**KEYWORDS** c-di-GMP, FleQ, transcriptome sequencing, LapA secretion, cAMP, adenylate cyclase

Cyclic diguanylate (c-di-GMP) is a ubiquitous bacterial second messenger that participates in regulation of a wide range of cellular processes through its downstream receptors (1). Several types of c-di-GMP receptors have been identified, including PilZ proteins, GGDEF, and/or EAL domain-containing proteins, riboswitches, and transcriptional regulators. Transcriptional regulatory c-di-GMP effectors modulate various bacterial physiological processes by directly regulating target gene expression in response to c-di-GMP. For example, transcriptional regulatory c-di-GMP effectors, such as VpsT

Address correspondence to Wenli Chen, wlchen@mail.hzau.edu.cn, or Qiaoyun Huang, qyhuang@mail.hzau.edu.cn.

of *Vibrio cholerae*, FleQ of *Pseudomonas aeruginosa*, and MrkH of *Klebsiella pneumoniae*, regulate transcription of biofilm matrix- and flagellum-related genes, thereby modulating biofilm formation and bacterial motility (2–4). The c-di-GMP-responsive regulator Clp from *Xanthomonas campestris* and the c-di-GMP-responsive regulator Bcam1349 from *Burkholderia cenocepacia* are involved in the regulation of bacterial virulence-related genes (5, 6). Two c-di-GMP-responsive regulators, LtmA and HpoR from *Mycobacterium smegmatis*, regulate lipid metabolism/transport and antioxidant defense, respectively (7, 8).

Among all transcriptional regulatory c-di-GMP effectors identified to date, FleQ is the best characterized in terms of its action mechanism. FleQ, as an NtrC subfamily regulator protein, is widely distributed in *Pseudomonas* species, and it contains an N-terminal atypical REC domain (also named FleQ domain), a central AAA+ ATPase domain, and a C-terminal helix-turn-helix DNA-binding domain (9). Transverse dimers formed via the atypical REC domain are essential for FleQ to function as a c-di-GMP receptor and flagellum gene regulator (10). FleQ was identified at first as a master regulator to activate transcription of flagellar genes (11, 12), and then it was found to bind c-di-GMP and regulate the genes involved in biofilm formation (3). By inversely regulating biofilm and flagellar gene expression, FleQ helps to control the plankton-to-biofilm lifestyle transition in response to c-di-GMP.

FleQ functions as both a repressor and an activator to bind to two sites on the promoter of the exopolysaccharide *pel* operon, and it controls the activity of the *pel* promoter along with FleN (another ATPase) in response to c-di-GMP in *P. aeruginosa* (13). Upon binding ATP, FleN forms dimers and interacts with the two FleQ molecules bound to DNA, and then the obtained FleQ-FleN-DNA complex further forms a bridge to inhibit transcription of the promoter, leading to *pel* repression. In the presence of c-di-GMP, FleQ undergoes a conformational change and then switches into an activator. Crystal structure analysis shows that c-di-GMP binding to the AAA+ ATPase domain of FleQ leads to ATPase active site obstruction, hexameric ring destabilization, and quaternary structure transition disruption, thereby altering the transcriptional activity of FleQ (14). A schematic diagram shows the mechanism by which c-di-GMP and FleQ/FleN coregulate target genes (Fig. 1).

Identifying new members of the c-di-GMP/FleQ regulon helps to enrich knowledge of FleQ. Through the search for FleQ binding consensus sequences in *Pseudomonas* genome sequences, several potential new target genes of FleQ have been obtained (15), such as the *siaABCD* operon and *bdlA* gene in *P. aeruginosa*, responsible for cell aggregation and biofilm dispersal, *lapA*-like adhesin genes, and a homologue of *gcbA* encoding diguanylate cyclase. Studies of *P. putida* confirm that *lapA* and *gcbA* are directly regulated by FleQ in response to c-di-GMP (16, 17). Chromatin immunoprecipitation sequencing (ChIP-seq) analysis reveals that FleQ regulates iron homeostasis-related genes in *P. fluorescens* and *P. putida* and that FleQ shares some common target genes with another global regulator, AmrZ, in *P. fluorescens*, especially iron-related genes, indicating cross talk between these two regulators (18).

To identify new target genes of c-di-GMP/FleQ, we carried out transcriptomic analysis to investigate the influence of high-level c-di-GMP and *fleQ* deletion on transcriptomic profiles of *P. putida*. By verifying the results from transcriptomic analysis, we discovered five new target genes under the direct regulation of c-di-GMP/FleQ. Further study concerning the functions of two target genes, *lapE* and *cyaA*, revealed the role of c-di-GMP/FleQ in LapA secretion and cyclic AMP (cAMP) synthesis regulation.

## RESULTS

**Identification of c-di-GMP-regulated genes through transcriptomic analysis.** WspR is a well-known diguanylate cyclase (DGC) with c-di-GMP-synthesizing activity in *Pseudomonas*, and a multicopy plasmid (pBBR1MCS5-*wspR*) containing *wspR* is used to increase intracellular c-di-GMP in wild-type KT2440, as previously reported (16, 19). The wild-type KT2440 harboring pBBR1MCS5-*wspR* is termed WT+*wspR*, and the wild-type

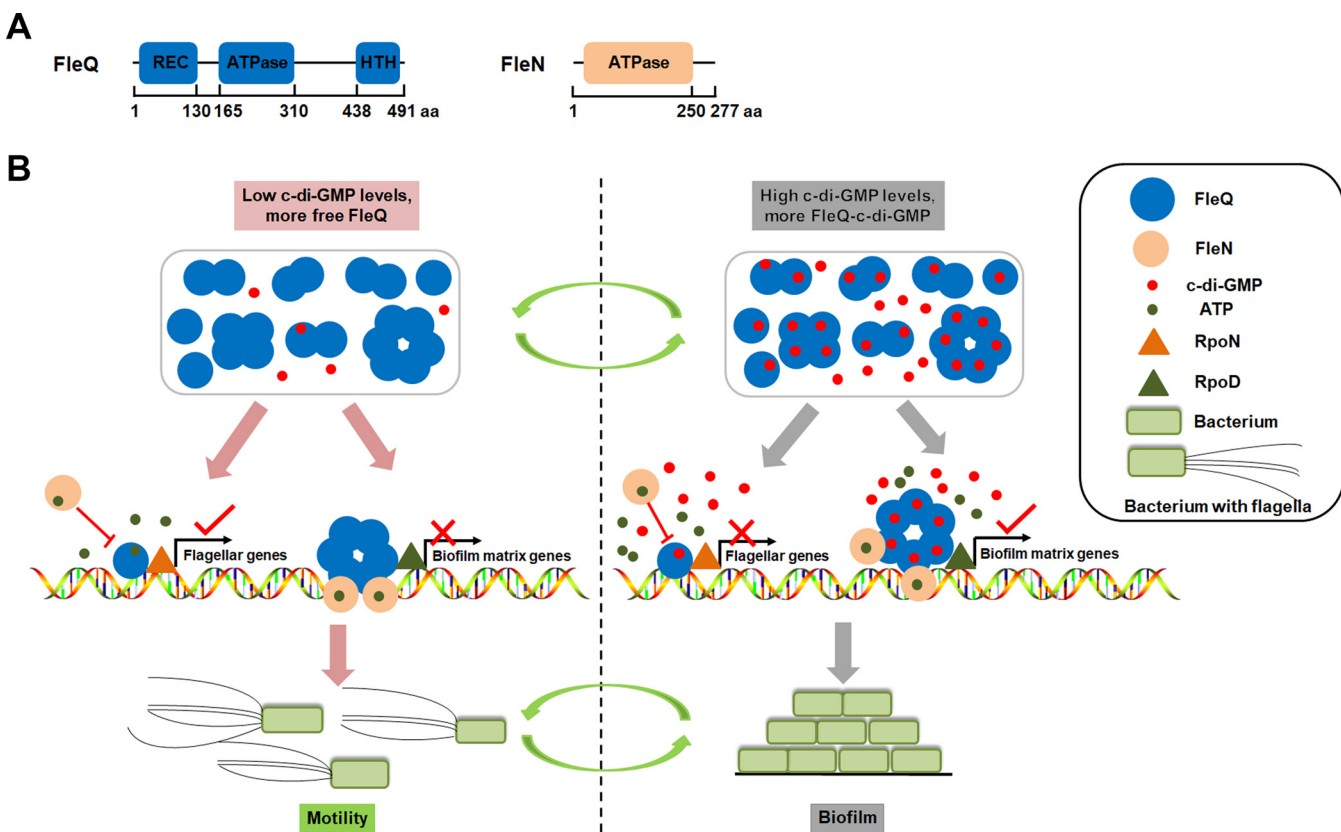

**FIG 1** Schematic diagram of the mechanism by which c-di-GMP and FleQ/FleN coregulate target genes. (A) Domain compositions of FleQ and FleN. The amino acid positions where the predicted domains start and end are shown. (B) Under low c-di-GMP levels, more non-c-di-GMP binding FleQ molecules (free FleQ) exist in the cell, and the free FleQ binds to promoters of flagellar genes (such as *fleS* and *fliF*). FleQ interacts with the sigma factor RpoN bound to the target promoter, this RpoN recruits RNA polymerase to the promoter, and then the ATPase domain of FleQ catalyzes ATP hydrolysis to form a RNA polymerase-promoter open complex and to initiate gene transcription. In the regulation of biofilm matrix genes (such as *pel* and *psl*), FleQ forms a hexamer and binds two sites of the promoter (one for repression and the other for activation), and then FleN hydrolyzes ATP to form dimers and subsequently interacts with FleQ molecules bound to the promoter to form a bridge, further inhibiting transcription. Under high c-di-GMP levels, more c-di-GMP-binding FleQ molecules exist, and such c-di-GMP binding inhibits its ATPase activity of FleQ, repressing transcription of flagellar genes. Meanwhile, c-di-GMP binding changes conformation of the FleQ hexamer and releases FleQ from the repression site of biofilm matix promoters. FleQ at the activation site activates the transcription together with another sigma factor, possibly RpoD.

KT2440 harboring empty plasmid pBBR1MCS5 is termed WT+control. p*CdrA*::*gfp*C-tet, a fluorescent reporter of c-di-GMP (20), was used to determine and compare the c-di-GMP levels in WT+control and WT+*wspR*. Normalized green fluorescent protein (GFP) fluorescence of WT+*wspR* was about threefold as high as that of WT+control (Fig. 2A), indicating that introducing pBBR1MCS5-*wspR* to the wild type provokes an increase in cellular c-di-GMP. To identify genes under the influence of c-di-GMP, transcriptomic analysis was performed to compare the transcriptome profiles of WT+*wspR* and WT+control with three technical replicates for each strain. A total of 283 differentially expressed genes (DEGs) were identified (fold change, ≥2 or ≤−2) under high c-di-GMP levels. Among them, 187 genes exhibited upregulated expression (see Table S1 in the supplemental material), and the remaining 96 genes showed downregulated expression (Table S2).

We summarized the numbers of DEGs involved in major physiological processes (Fig. 2B). The 48 DEGs were found to be related to cell motility and secretion processes, which were the most well-known processes that c-di-GMP was involved in. Meanwhile, 39 DEGs were related to amino acid transport/metabolism processes, 34 DEGs to carbohydrate transport/metabolism process, and 20 DEGs to secretion and transport processes, and these were relatively less reported processes that c-di-GMP was involved in. In addition, 68 DEGs encode hypothetical proteins of unknown function.

**Identification of FleQ-regulated genes through transcriptomic analysis.** To identify the genes under the influence of FleQ, transcriptomic analysis was performed

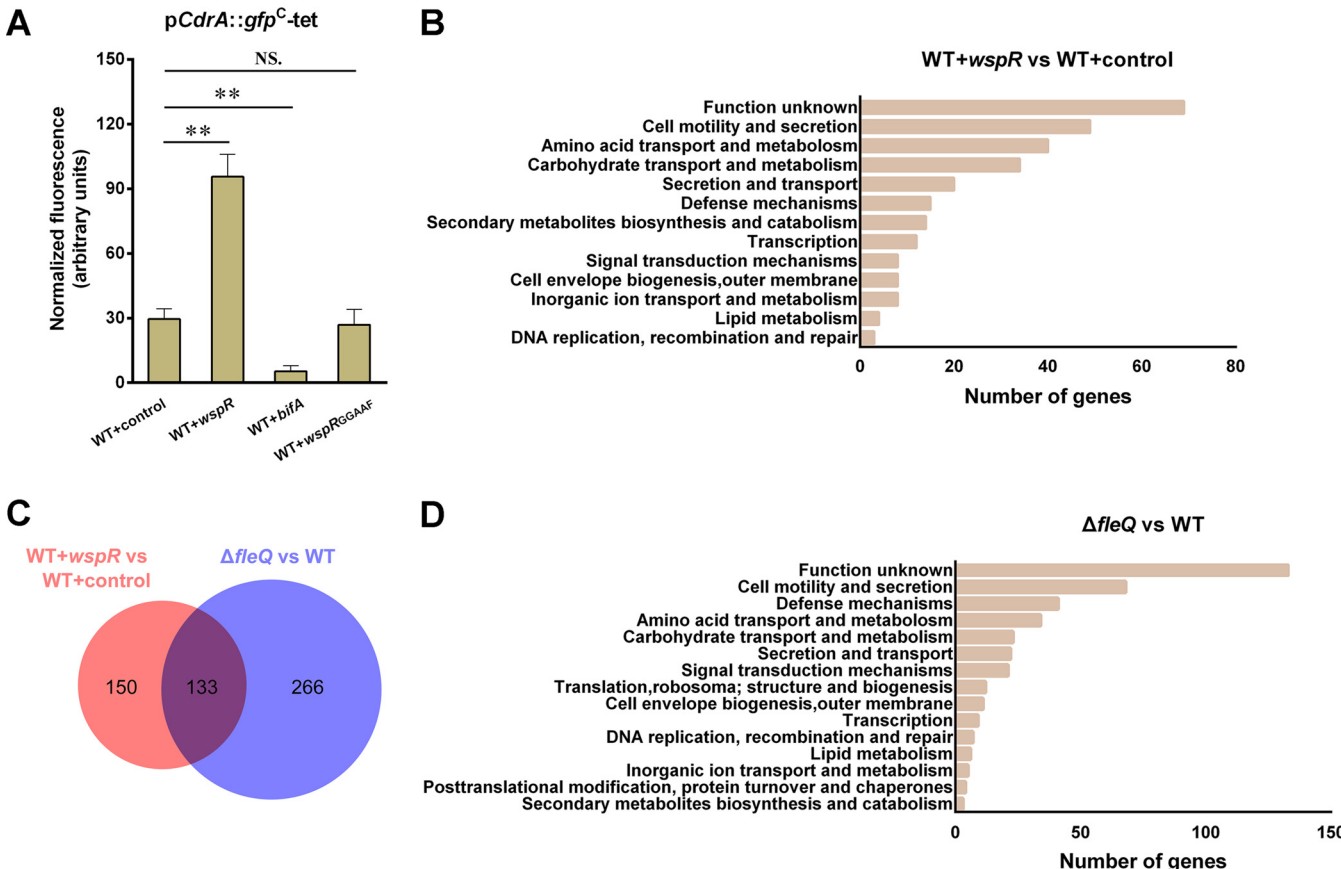

**FIG 2** Transcriptomic analysis of genes under influence of high-level c-di-GMP and FleQ. (A) Effect of multicopy *wspR*, *bifA*, or mutated *wspR* on the intracellular c-di-GMP level. Plasmid p*CdrA::gfpC-tet* was used to determine c-di-GMP levels. GFP fluorescence of stationary-phase (24 h) LB cultures was measured. Results are averages from three independent assays. The values represent mean values with standard deviations (**, $P \leq 0.01$). (B and D) The numbers of genes involved in major physiological processes under the influence of high c-di-GMP levels (B) and *fleQ* deletion (D) at the transcriptional level in *P. putida* KT2440. Information on gene functions and physiological processes can obtained from the annotated genome database at http://www.pseudomonas.com (48). (C) Venn diagram illustrating numbers of differentially expressed genes (DEGs) in WT+*wspR* and *fleQ* deletion mutant. The red circle represents DEGs identified from WT+*wspR* relative to WT+control, and the blue circle represents DEGs identified from the *fleQ* mutant relative to WT+control. The numbers of DEGs in WT+*wspR* alone (150), *fleQ* mutant alone (266), and the common DEGs shared by WT+*wspR* and *fleQ* mutant (133) are shown in the diagram.

to compare the transcriptome profiles of an *fleQ* deletion mutant (Δ*fleQ*) and wild-type KT2440 with three technical replicates for each strain. A total of 399 DEGs were identified (fold change, $\geq 2$ or $\leq -2$) in the *fleQ* mutant, of which 172 genes showed upregulated expression (Table S3), and the remaining 227 genes showed downregulated expression (Table S4). Of the 399 DEGs, 68 DEGs were related to cell motility and secretion processes, which were the most well-reported processes that FleQ was involved in (Fig. 2D). Some DEGs were found to be related to defense mechanisms (41 DEGs), amino acid transport/metabolism (34 DEGs), carbohydrate transport/metabolism (23 DEGs), and signal transduction (21 DEGs). Meanwhile, 133 DEGs were responsible for encoding putative function-unknown proteins.

**Identification of c-di-GMP/FleQ-regulated genes.** c-di-GMP binds to FleQ to modulate gene transcription (3, 13, 14). Theoretically, genes regulated by FleQ are also influenced by c-di-GMP, but genes regulated by c-di-GMP are not necessarily influenced by FleQ, since there are other potential c-di-GMP-responsive transcriptional regulators. The purpose of this study was to identify new target genes coregulated by c-di-GMP and FleQ. The first transcriptomic analysis (WT+*wspR* versus WT+control) described above identified the potential genes regulated by c-di-GMP, and the second transcriptomic analysis (Δ*fleQ* versus WT) identified the potential genes regulated by FleQ. Thus, the genes coregulated by c-di-GMP and FleQ should be found in both the first and the second transcriptomic analysis. Considering this, we matched the 283

mSystems®

DEGs identified in WT+*wspR* with the 399 DEGs identified in the *fleQ* mutant and found 133 common DEGs (Fig. 2C). Of these 133 DEGs, 47 DEGs were reported to be coregulated by c-di-GMP and FleQ, including 1 *lapA* gene, 7 *bcs* genes, and 39 flagellum- and motility-related genes (3, 11, 12, 16), and they were not investigated in the following studies. The remaining 86 DEGs were potentially new target genes of c-di-GMP/FleQ (Table 1), and they belonged to 68 operons in terms of their distribution in the KT2440 genome (GenBank accession no. NC_002947.3).

To verify the results of transcriptomic analysis, we compared transcriptions of these 68 operons in WT+*wspR*, *fleQ* mutant harboring control vector (△*fleQ*+control), and WT+control using quantitative PCR (qRT-PCR). One gene was chosen from each operon to test. Transcription levels of 50 out of the 68 genes exhibited significant differences between WT+control and WT+*wspR* or △*fleQ*+control (Fig. 3A and B), and these 50 genes exhibited transcription change trends similar to those obtained from the transcriptomic analysis. The qRT-PCR assay results indicated that transcription levels of the remaining 18 genes showed no obvious difference between WT+control and WT+*wspR* or △*fleQ*+control, suggesting that these 18 genes were not regulated by c-di-GMP/FleQ, and transcriptomic analysis results of these genes and related operons were false positive. Taken together, 50 new operons under the influence of c-di-GMP/FleQ were identified.

**FleQ specifically binds with the upstream sequences of *PP_0681*, *PP_0788*, *lapE*, *cyaA*, and *PP_5586*.** FleQ regulates transcription of target genes by directly binding to their promoters (11, 15). To determine those operons directly regulated by FleQ from the 50 identified operons, we carried out electrophoretic mobility shift assays (EMSAs) to test whether FleQ could bind upstream sequence of each of these 50 operons and found that 7 upstream sequence fragments, including *PP_0681*, *PP_0788*, *PP_4519*(*lapE*)p, *PP_4858*, *PP_5222*(*cyaA*)p, *PP_5496*, and *PP_5586*, exhibited a stepwise increase in the shifted DNA amount, with FleQ protein amount increasing from 100 nM to 300 nM (Fig. 4A). Binding of FleQ with the seven labeled DNAs was not disrupted by a pUC19 fragment (unlabeled nontarget DNA) (Fig. 4A). EMSA results indicated that the band shifts of *PP_0681*, *PP_0788*, *lapE*p, *cyaA*p, and *PP_5586* were stronger than those of *PP_4858* and *PP_5496* and that the remaining 43 promoters showed no band shift (Fig. S1). These results indicated that FleQ bound to upstream sequences of *PP_0681*, *PP_0788*, *lapE*, *PP_4858*, *cyaA*, *PP_5496*, and *PP_5586* in *in vitro* EMSA.

In previous studies, a bacterial one-hybrid assay (B1H) was applied to detect protein–DNA interactions *in vivo* using a reporter plasmid, pBXcmT, containing the selectable genes *HIS3* and *aadA* and a plasmid, pTRG, harboring a gene encoding the alpha subunit of RNA polymerase. The regulator protein has been reported to be able to recruit RNA polymerase and stabilize its binding to the promoter on pBXcmT, activating the transcription of *HIS3* and *aadA*, so that the reporter strains could survive on screening medium containing 3-amino-1,2,4-triazole (3-AT) and streptomycin only when this regulator protein interacts with the promoter on pBXcmT (21). Based on these findings, we performed B1H to verify our EMSA results. As shown in Fig. 4B, the reporter strains containing both pTRG-*fleQ* and pBXcmT with any of the five promoters (*PP_0681*, *PP_0788*, *lapE*p, *cyaA*p, and *PP_5586*) grew well, but the strain containing pTRG-*fleQ* and pBX-*PP_4858* or pBX-*PP_5496* exhibited no growth on the screening medium. The self-activation control strain containing either pTRG-*fleQ* or reporter plasmid alone also did not grow on the screening medium. These results indicated that FleQ specifically bound to the upstream sequences of *PP_0681*, *PP_0788*, *lapE*, *cyaA*, and *PP_5586*.

To determine the precise binding site of FleQ in the five promoters, a DNase I footprinting assay was performed. The results indicated that TGAGTCAATAAACTGGCGCTG (−165 to −145 bp relative to the translational start site) sequence in *PP_0681*, AACGGCGCTGG (−113 to −103 bp relative to the translational start site) and TATTTGGCGTCATAG (−67 to −53 bp relative to the translational start site) sequences in *PP_0788*, CAAAGTGACAATATTTTGTCGCCAA sequence (−319 to −295 bp relative to the translational start site) in *lapE*p, and TGCATAATCTGCATGTCGT (−175 to −157 bp relative

**TABLE 1** Genes exhibiting significant differences in transcription levels in both WT+*wspR* versus WT+control and Δ*fleQ*+control versus WT+control[a]

| Gene ID | Fold change in: | | Gene name | Description |
| | WT+*wspR* versus WT+control | Δ*fleQ* mutant versus WT | | |
| --- | --- | --- | --- | --- |
| PP_0089 | 2.081 | 7.48 | *osmC* | Stress-induced peroxiredoxin |
| PP_0115 | 2.244 | 10.86 | *katE* | Hydroperoxidase |
| PP_0817 | −4.499 | −2.623 | *alaC* | Aminotransferase |
| PP_1502 | 2.414 | 6.765 | | OmpA family protein |
| PP_1895 | 2.896 | 5.952 | *yadG* | ABC transporter ATP-binding protein |
| PP_1896 | 2.553 | 2.928 | *yadH* | ABC transporter permease |
| PP_1970 | −6.342 | −9.33 | | Lipoprotein |
| PP_2125 | 2.2 | 2.811 | *yegS* | Lipid kinase |
| PP_2358 | −2.259 | −2.193 | | Putative type 1 pilus subunit CsuA/B protein |
| PP_2359 | −2.406 | −2.078 | | Putative type 1 pilus subunit CsuA/B protein |
| PP_2360 | −2.22 | −2.457 | | Type I pilus subunit CsuA/B |
| PP_2362 | −2.104 | −2.303 | | Usher protein |
| PP_2561 | 2.777 | 5.809 | | Hemolysin-type calcium-binding bacteriocin |
| PP_2647 | 30.251 | 32.067 | | MFS transporter |
| PP_2689 | 2.354 | 2.221 | | Endoribonuclease |
| PP_2827 | 11.051 | 14.62 | | Alcohol dehydrogenase |
| PP_2914 | 2.057 | 3.308 | *proP* | Osmosensory proline/betaine/H$^+$ permease |
| PP_3089 | −2.04 | 2.717 | *hcp1* | Hcp1 |
| PP_3096 | −2.006 | 2.494 | *tssG1* | TssG1 |
| PP_3097 | −2.407 | 2.535 | *tssF1* | TssF1 |
| PP_3100 | −2.128 | 2.833 | *tssB1* | TssB1 |
| PP_3260 | 2.242 | 8.681 | *ligD* | DNA ligase D |
| PP_3360 | 2.218 | 6.175 | | Membrane protein |
| PP_3425 | 1073.048 | 897.89 | | RND family transporter MFP subunit |
| PP_3426 | 329.193 | 276.409 | *mexF* | Multidrug RND transporter MexF |
| PP_3427 | 205.080 | 159.445 | *oprN* | Multidrug RND transporter outer membrane protein OprN |
| PP_3455 | −2.714 | −2.606 | | Multidrug RND transporter membrane fusion protein |
| PP_3456 | −2.179 | −4.668 | *mexB* | Multidrug resistance protein MexB |
| PP_3503 | 6.009 | 3.556 | | Sigma-54 dependent transcriptional regulator |
| PP_3519 | 32.250 | 21.445 | | Lipoprotein |
| PP_3541 | 2.027 | 2.404 | | MgtC family transporter |
| PP_3613 | 2.069 | 4.476 | | L-Sorbosone dehydrogenase |
| PP_3878 | 2.257 | −2.87 | | Minor capsid protein C |
| PP_3941 | 3.585 | 2.758 | *nicF* | Maleamate amidohydrolase |
| PP_3942 | 3.891 | 3.150 | *nicE* | Maleate isomerase |
| PP_3943 | 3.983 | 2.488 | *nicD* | N-formylmaleamate deformylase |
| PP_3944 | 3.989 | 3.070 | *nicC* | 6-hydroxynicotinate 3-monooxygenase |
| PP_3945 | 2.667 | 2.918 | *nicX* | 2,5-dihydroxypyridine 5,6-dioxygenase |
| PP_4057 | 7.307 | 5.311 | | membrane protein |
| PP_4434 | 5.950 | 2.479 | *dadAI* | D-Amino acid dehydrogenase small subunit |
| PP_4519 | 2.893 | 7.296 | *tolC* | Agglutination protein |
| PP_4856 | 2.167 | 6.544 | | Dps family ferritin |
| PP_4983 | 2.550 | 3.045 | | Amine oxidase |
| PP_5033 | 41.840 | −2.460 | *hutU* | Urocanate hydratase |
| PP_5222 | −2.256 | 3.010 | *cyaA* | Adenylate cyclase |
| PP_5269 | 11.139 | 4.010 | *dadX* | Alanine racemase |
| PP_5270 | 5.722 | 3.509 | *dadAII* | D-Amino acid:quinone oxidoreductase |
| PP_5298 | −2.615 | −2.316 | | Glutamine amidotransferase |
| PP_5299 | −2.752 | −2.209 | *puuAII* | Glutamate-putrescine ligase |
| PP_0584 | −2.54 | −4.18 | | Methyl-accepting chemotaxis transducer |
| PP_1371 | −3.79 | −16.49 | *pctA* | Methyl-accepting chemotaxis protein PctA |
| PP_1819 | −2.426 | −2.632 | | Methyl-accepting chemotaxis transducer |
| PP_2249 | −4.663 | −11.056 | *pctB* | Methyl-accepting chemotaxis protein PctB |
| PP_3557 | −2.710 | −3.439 | | methyl-accepting chemotaxis transducer |
| PP_4888 | −2.570 | −4.793 | | Methyl-accepting chemotaxis transducer |
| PP_5020 | −4.702 | −13.313 | | Methyl-accepting chemotaxis protein |
| PP_0681 | 3.128 | 2.467 | | Hypothetical protein |
| PP_0788 | −11.795 | 6.391 | | Hypothetical protein |
| PP_1503 | 2.555 | 6.825 | | Hypothetical protein |
| PP_1691 | 4.986 | 2.156 | | Hypothetical protein |

**TABLE 1** (Continued)

| Gene ID | Fold change in: | | Gene name | Description |
|---|---|---|---|---|
| | WT+*wspR* versus WT+control | Δ*fleQ* mutant versus WT | | |
| PP_1828 | −3.956 | −16.106 | | Hypothetical protein |
| PP_2059 | 2.102 | 4.908 | | Hypothetical protein |
| PP_2858 | 29.15 | −7.298 | | Hypothetical protein |
| PP_3104 | −2.238 | 3.096 | | Hypothetical protein |
| PP_3261 | 2.523 | 13.247 | | Hypothetical protein |
| PP_3524 | 2.007 | 5.944 | | Hypothetical protein |
| PP_3542 | 2.092 | 2.267 | | Hypothetical protein |
| PP_3770 | 53.334 | 48.34 | | Hypothetical protein |
| PP_3795 | −3.555 | −6.312 | | Hypothetical protein |
| PP_3855 | 3.297 | −2.668 | | Hypothetical protein |
| PP_3856 | 2.217 | −2.995 | | Hypothetical protein |
| PP_3874 | 2.327 | −3.28 | | Hypothetical protein |
| PP_3928 | 5.593 | 10.226 | | Hypothetical protein |
| PP_4406 | −4.688 | −8.016 | | Hypothetical protein |
| PP_4858 | 70.390 | 42.431 | | Hypothetical protein |
| PP_5073 | −5.853 | −3.019 | | Hypothetical protein |
| PP_5430 | 3.929 | −6.020 | | Hypothetical protein |
| PP_5462 | 2.957 | −2.611 | | Hypothetical protein |
| PP_5496 | 79.486 | 44.679 | | Hypothetical protein |
| PP_5524 | 2.395 | 23.189 | | Hypothetical protein |
| PP_5542 | 2.816 | 3.496 | | Hypothetical protein |
| PP_5549 | 4.553 | −2.003 | | Hypothetical protein |
| PP_5560 | 2.394 | 16.782 | | Hypothetical protein |
| PP_5586 | 80.559 | 3.473 | | Hypothetical protein |
| PP_5592 | 3.287 | 8.070 | | Hypothetical protein |
| PP_5710 | −2.442 | −39.200 | | Hypothetical protein |

[a]Positive values indicate the increased transcription levels, and negative values indicate the decreased transcription levels. Detailed information of gene names and descriptions can be obtained from the annotated genome database at http://www.pseudomonas.com (48).

to the translational start site) and TATGGTGTCGGATCATTGA (−118 to −100 bp relative to the translational start site) sequences in *PP_5586* were protected by FleQ protein (Fig. 4C). Unfortunately, we tried several times but failed to obtain the precise binding site(s) in the DNase I footprinting assay of the *cyaA* promoter. To locate the specific region of the *cyaA* promoter interacting with FleQ, three truncated fragments of *cyaA* promoters were amplified for EMSA. As shown in Fig. 4D, *cyaA*pF2 and *cyaA*pF3 DNA fragments produced a band shift on the gel, whereas no band shift was observed with *cyaA*pF1, indicating that the binding site(s) is located between positions −139 and −51 on the *cyaA* promoter relative to its translational start site.

Since c-di-GMP can change the oligomerization of FleQ and, thus, change FleQ/promoter binding (13, 14), we added c-di-GMP (80 μM) to the reaction solution to test the effect of c-di-GMP on the binding of FleQ to target promoters in a DNase I footprinting assay. However, we failed to observe any change in the binding sites with or without c-di-GMP (data not shown). We then added c-di-GMP (from 0 to 90 μM) to the reaction solution in EMSA. The EMSA results showed that c-di-GMP enhanced the binding of FleQ to *PP_0681*pro, *lapE*pro, and *PP_5586*pro, but it had no obvious influence on the binding of FleQ to *PP_0788*pro and *cyaA*pro (Fig. 4E).

**c-di-GMP regulates expression of *PP_0681*, *PP_0788*, *lapE*, *cyaA*, and *PP_5586* in FleQ-dependent manner.** To further determine whether c-di-GMP regulated the expression of the five new target genes via FleQ, we compared promoter activities of the five genes under different c-di-GMP levels in wild-type KT2440 and the *fleQ* mutant by using β-galactosidase (LacZ) promoter fusion reporters. High c-di-GMP levels were achieved by expressing WspR as described above, and low c-di-GMP levels were obtained by expressing the phosphodiesterase BifA as previously reported (Fig. 2A) (16). Promoter activities of *PP_0681*, *PP_5586*, and *lapE* were decreased in WT+*bifA* and increased in WT+*wspR* compared with those in WT+control, but promoter activities of *PP_0788* and *cyaA* showed an opposite trend. In the *fleQ* deletion mutant, the promoter activity of each gene showed no obvious differences between Δ*fleQ*+*wspR* or

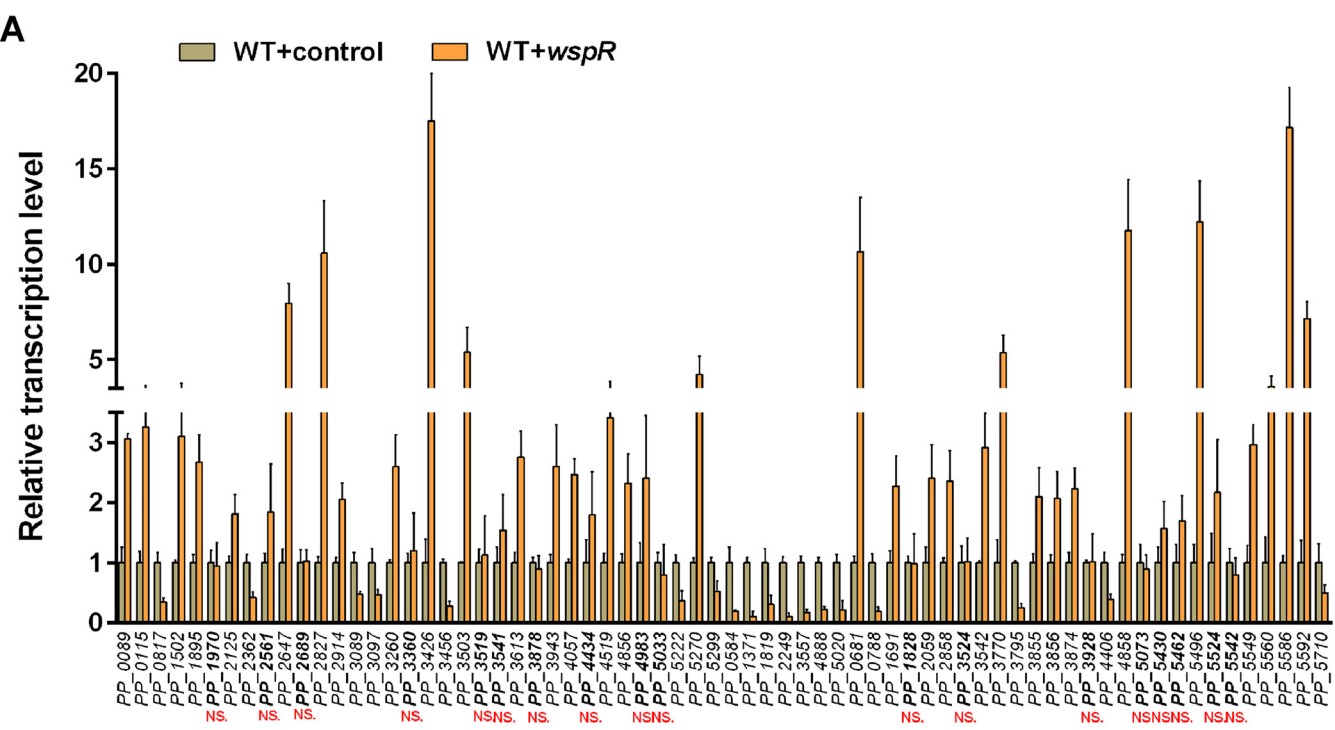

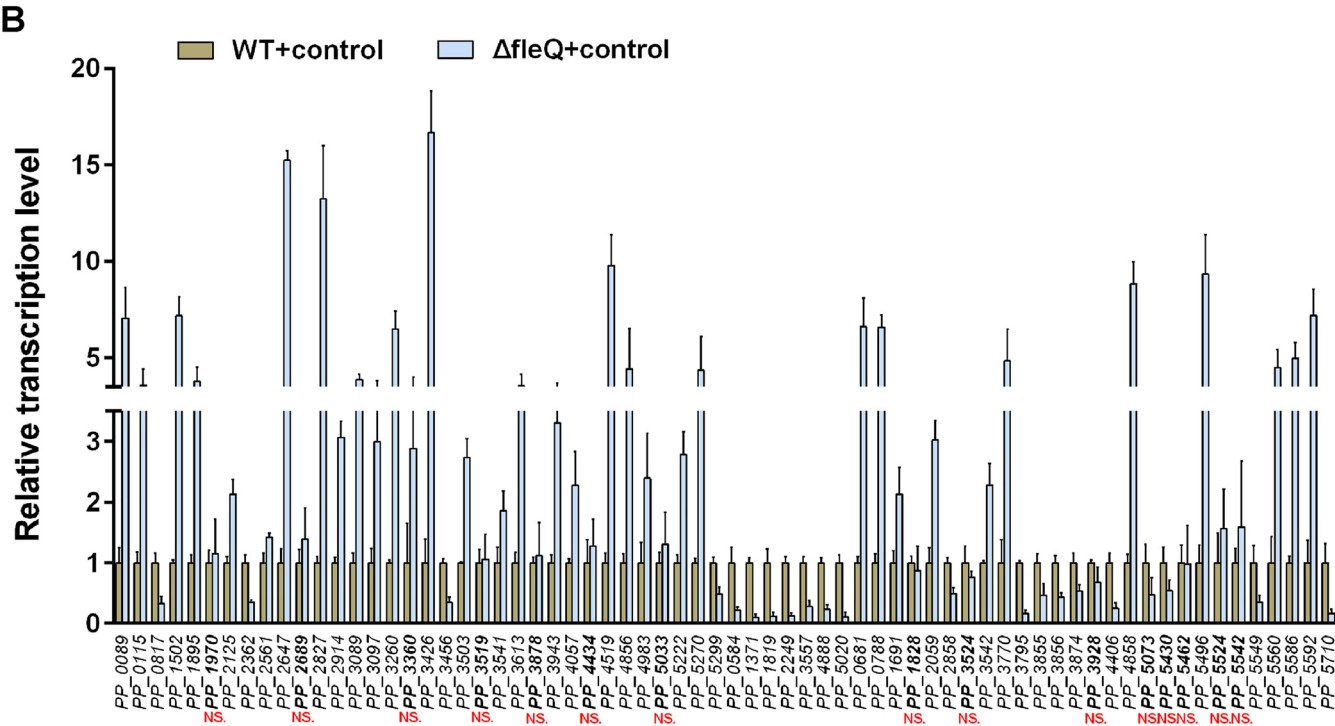

**FIG 3** Comparison of relative transcriptional levels of 68 operons between WT+control and WT+*wspR* (A) and between WT+control and Δ*fleQ*+control (B) by qRT-PCR. Transcriptional levels in WT+control are used as a reference. The results are averages from three independent assays. The data values represent mean values with standard deviations for three biologically independent samples. NS, no statistically significant difference in transcription level between WT+control and WT+*wspR* (A) or between WT+control and Δ*fleQ*+control (B).

Δ*fleQ*+*bifA* and Δ*fleQ*+control, indicating that the regulation of target genes by c-di-GMP was abolished (Fig. 5). These results suggested that c-di-GMP regulated expression of *PP_0681*, *PP_0788*, *lapE*, *cyaA*, and *PP_5586* in a FleQ-dependent manner.

In this study, the overexpression of WspR resulted in high-level c-di-GMP, further changing the transcription of target genes, but transcription changes possibly were

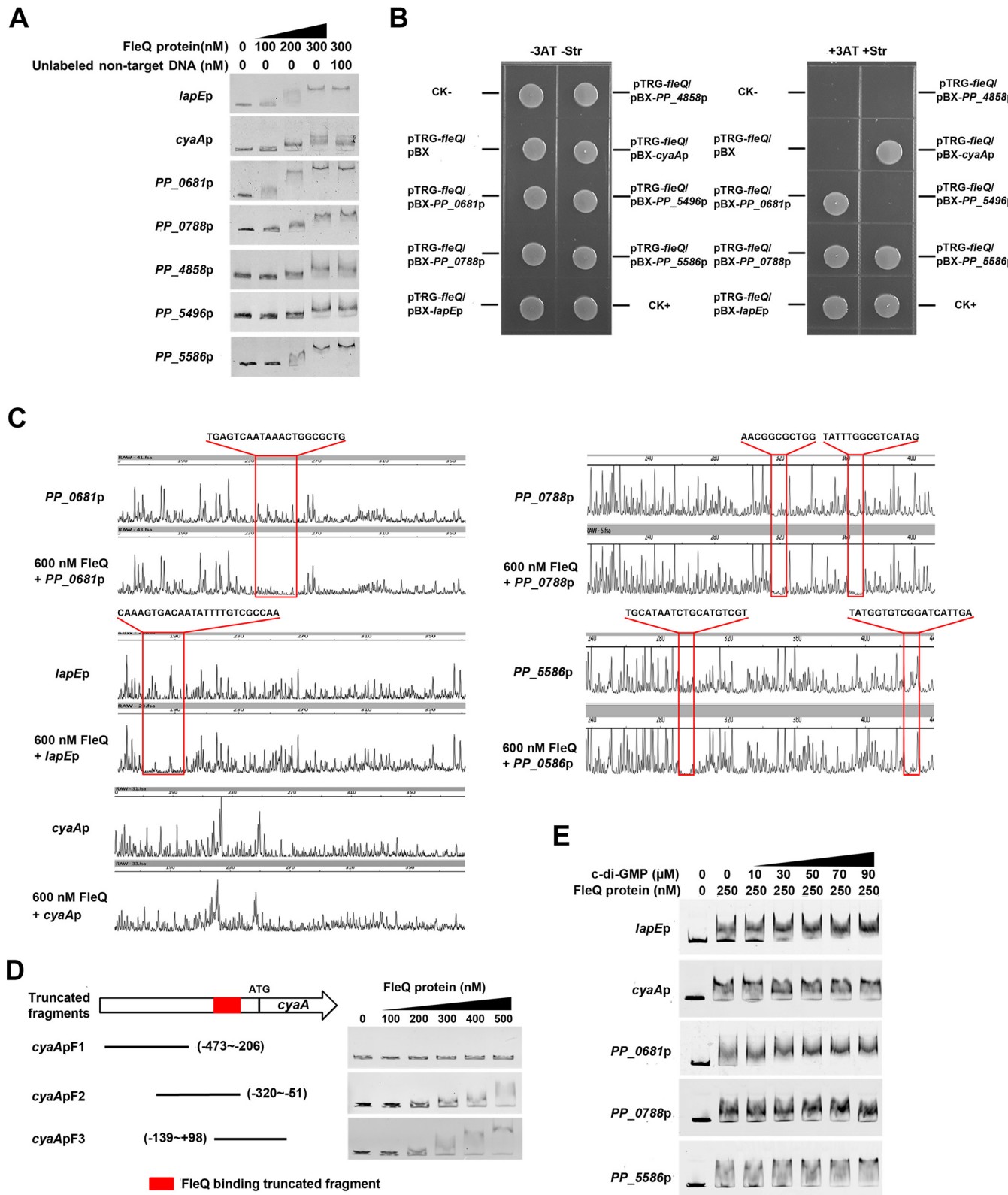

**FIG 4** Analysis for interactions between FleQ and upstream sequences of *PP_0681*, *PP_0788*, *lapE*, *cyaA*, and *PP_5586*. (A) FleQ protein binds to upstream sequences of the seven genes in EMSA. The concentrations of FleQ and the amounts of unlabeled DNA used are shown. Unlabeled pUC19 fragment was used for competition experiments. (B) Bacterial one-hybrid assays of the interactions between FleQ and promoters of the seven target genes. Cotransformants containing plasmids pTRG-Rv3133+pBX-Rv2031 and pTRG+pBX were used as positive and negative controls, respectively. (C) DNase I footprinting assay with fragments containing the promoters of *PP_0681*, *PP_0788*, *lapE*, *cyaA*, and *PP_5586* in the presence and absence of FleQ. The protected regions and sequences are boxed. (D) Binding of truncated *cyaA* promoter fragments to FleQ. Schematic

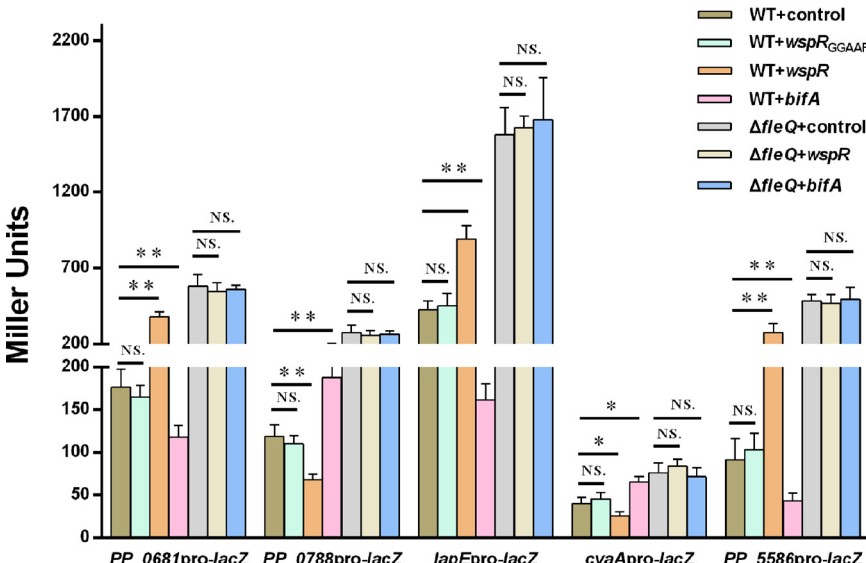

**FIG 5** Influence of c-di-GMP on promoter activities of target genes in wild type and *fleQ* deletion mutant. Promoter activities were analyzed by using promoter-*lacZ* fusion reporter plasmids. Strains containing control vector or *wspR*<sub>GGAAF</sub> represent normal c-di-GMP levels, strains containing multiple *wspR* represent high c-di-GMP levels, and strains containing multiple *bifA* represent low c-di-GMP levels. LacZ activity was measured from stationary-phase (24 h) LB cultures by following the method described in Materials and Methods. The values represent mean values with standard deviations for three biologically independent samples (**, $P \leq 0.01$; *, $P \leq 0.05$). NS, not statistically significant between two compared strains.

caused by unintended effects of overexpressing a protein. To test this possibility, we introduced a point mutation of the GGDEF motif of WspR (from GGEEF to GGAAF) to abolish the c-di-GMP synthesis ability of WspR. Western blot results showed that the point-mutated WspR was detected in both the wild-type strain and *fleQ* mutant with their molecular weight and signal intensity similar to those of the wild-type WspR, indicating that the point-mutated WspR was stable in both the wild type and *fleQ* mutant (Fig. S2). Normalized GFP fluorescence of the c-di-GMP reporter exhibited no obvious difference between WT+*wspR*<sub>GGAAF</sub> and WT+control (Fig. 2A), indicating that introduction of the point-mutated WspR to the wild type had no influence on cellular c-di-GMP. The influence of the point-mutated WspR on activities of the five target promoters was investigated. The results showed that all five promoters showed no obvious difference in activity between WT+control and WT+*wspR*<sub>GGAAF</sub> (Fig. 5), implying that the transcriptional changes were caused by c-di-GMP rather than by unintended effects of overexpression of WspR.

**ATPase activity, RpoN binding ability, and c-di-GMP binding ability of FleQ are not required for the complementation of new target genes.** Typically, FleQ interacts with the sigma factor RpoN bound to the promoter of the flagellar gene. This RpoN recruits RNA polymerase to the promoter, and then the ATPase domain of FleQ catalyzes ATP hydrolysis to form an RNA polymerase-promoter open complex and to initiate gene transcription (11). However, ATPase activity and RpoN binding ability of FleQ are not required for regulation of exopolysaccharide-synthesizing operons (15, 17). The key roles of mutants of amino acid residues T224 and D245 in abolishing RpoN binding ability and ATPase activity of FleQ are confirmed from *P. aeruginosa* (22). To check whether the positions of these amino acid residues were the same in *P. aeruginosa* and

**FIG 4** Legend (Continued)

diagram of truncated fragments of *cyaA* promoter DNA is shown on the left. EMSA results are shown on the right. The coordinates represent the position relative to the ATG initiation codon. Red boxes represent the position of the truncated fragments bound to FleQ. (E) Influence of c-di-GMP on binding of FleQ to upstream sequences of the five target genes in EMSA. Concentrations of FleQ and c-di-GMP used are shown above each lane.

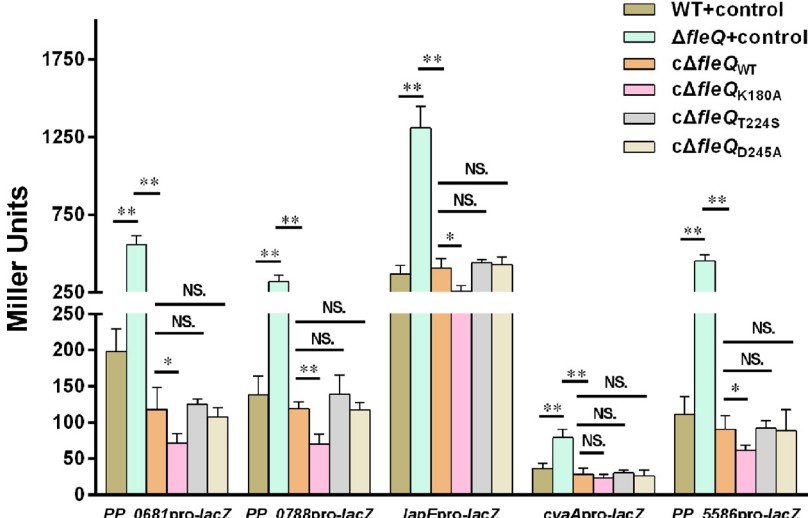

**FIG 6** Influence of FleQ point mutation on promoter activities of target genes. cΔfleQ$_{WT}$ corresponds to the *fleQ* mutant complemented with a plasmid carrying wild-type *fleQ*, and cΔfleQ$_{K180A/T224S/D245A}$ corresponds to the *fleQ* mutant complemented with a plasmid carrying point-mutated *fleQ*. Two asterisks above the column represent statistically significant differences. NS, not statistically significant between two compared strains. The values represent mean values with standard deviations for three biologically independent samples (**, $P \leq 0.01$).

*P. putida*, we aligned the amino acid sequence of FleQ in *P. putida* KT2440 with that in *P. aeruginosa* PAO1. Results showed that FleQ in *P. aeruginosa* and FleQ in *P. putida* shared 84% amino acid sequence identity, and that positions of the two amino acid residues (T224 and D245) were the same in the two bacteria (Fig. S3). Based on this, the mutants FleQ$_{T224S}$ and FleQ$_{D245A}$ were separately introduced to construct the *fleQ* complementation plasmids, and the obtained complementation plasmids were transformed into *fleQ* mutant to obtain a complementation strain (termed cΔfleQ$_{T224S/D245A}$). Promoter activities of the five target genes in cΔfleQ$_{T224S/D245A}$ were tested and compared with those in the wild-type strain, *fleQ* mutant, and *fleQ* mutant complemented with wild-type FleQ (termed cΔfleQ$_{WT}$). The results showed that transcriptional activities of all five target promoters were totally restored by the two point-mutated FleQ proteins to wild-type FleQ, indicating that point mutations of FleQ$_{T224S}$ and FleQ$_{D245A}$ had no obvious influence on complementation of all five new target genes (Fig. 6). These results suggested that neither the ATPase activity nor RpoN binding ability of FleQ were required for complementation of *PP_0681*, *PP_0788*, *lapE*, *cyaA*, and *PP_5586*.

Mutation of another conserved amino acid residue (K180A) in the ATPase domain of FleQ has been reported to abolish both c-di-GMP binding ability and ATPase activity of FleQ (22). Based on this finding, we introduced mutant FleQ$_{K180A}$ to construct the *fleQ* complementation plasmid and transformed this complementation plasmid into the *fleQ* mutant to obtain a complementation strain (termed cΔfleQ$_{K180A}$). Our data showed that the promoter activities of all five target genes were decreased in cΔfleQ$_{K180A}$ compared with those in the *fleQ* mutant, indicating that the point-mutated FleQ$_{K180A}$ could restore promoter activities of these target genes (Fig. 6). Our data also showed that the promoter activities of the four target genes (*PP_0681*pro, *PP_0788*pro, *lapE*pro, and *PP_5586*pro) in cΔfleQ$_{K180A}$ were lower than those in cΔfleQ$_{WT}$, indicating that FleQ$_{K180A}$ showed better repression ability than FleQ$_{WT}$ for these four target genes. Promoter activity of *cyaA* was not influenced by mutant FleQ$_{K180A}$, and FleQ$_{WT}$ and FleQ$_{K180A}$ showed similar repression of activity of *cyaA*pro (Fig. 6), suggesting that the impact of FleQ on *cyaA* was distinct from that on the other genes. These results demonstrated that the c-di-GMP-binding ability of FleQ was not required for complementation of *PP_0681*, *PP_0788*, *lapE*, *cyaA*, and *PP_5586*.

**Transcriptional changes of *lapE* and *cyaA* lead to corresponding protein level changes.** c-di-GMP/FleQ modulates the lifestyle transition from plankton to biofilm by regulating expression of biofilm- and flagellum-related genes (3, 15, 16). In this study, we identified five target genes coregulated by FleQ and c-di-GMP in *P. putida* through transcriptomics and protein-DNA binding assays. c-di-GMP/FleQ inhibited the transcription of *PP_0788* and *cyaA* and promotes the transcription of *PP_0681*, *lapE*, and *PP_5586*. We further investigated the functions of the five genes in *P. putida* and the effect of c-di-GMP/FleQ on the phenotypes related to these genes. BLAST results revealed that *PP_0681*, *PP_0788*, and *PP_5586* encoded putative function-unknown proteins but *lapE* and *cyaA* did not; thus, *lapE* and *cyaA* were further investigated.

Before we examined the influence of c-di-GMP/FleQ on the phenotypes related to *lapE* and *cyaA*, we first tested whether the transcriptional changes of *lapE* and *cyaA* lead to corresponding protein level changes. We fused the *gfp* gene to the end of *lapE-cyaA* to achieve fusion expression in a plasmid, and transcriptions of *lapE* and *cyaA* from the plasmid were under the control of their native promoters. We then transformed the plasmids into the wild-type strain, *bifA* mutant (Δ*bifA*), *fleQ* mutant (Δ*fleQ*), and *fleQ-bifA* double mutant (Δ*fleQ* Δ*bifA*) and further tested their GFP fluorescence intensity. Our previous study has shown that deletion of BifA (a phosphodiesterase with c-di-GMP degradation activity) causes increased c-di-GMP levels in *P. putida* (23). The results showed that GFP fluorescence intensity of the Δ*bifA* mutant containing the *lapE-gfp* fusion reporter was much higher than that of the wild type, whereas the fluorescence intensity of the Δ*bifA* mutant containing the *cyaA-gfp* fusion reporter was lower than that of the wild type (Fig. 7A). Fluorescence intensity of the Δ*fleQ* mutant containing either fusion reporter was higher than that of the wild type. The Δ*fleQ* Δ*bifA* mutant containing either fusion reporter showed fluorescence intensity similar to that of the Δ*fleQ* mutant, suggesting that the modulation of c-di-GMP on LapE and CyaA protein levels was FleQ dependent. These results indicated that under high c-di-GMP levels, transcriptional levels of *lapE* and *cyaA* affected the corresponding protein levels in an FleQ-dependent manner.

**LapE is responsible for LapA secretion and biofilm formation.** LapE (Pfl01_1462) in *P. fluorescens* Pf0-1, which shares about 73.3% amino acid sequence identity with LapE of *P. putida*, is responsible for secretion of LapA (an adhesin), which plays a vital role in initial attachment and biofilm formation (24–26). To test the role of LapE in LapA secretion and biofilm formation in *P. putida*, a *lapE* deletion mutant (Δ*lapE*) was constructed using wild-type KT2440 containing a 3× hemagglutinin (HA) tag in LapA. Biofilm formation and LapA secretion of the mutant were assessed. Deletion of *lapE* abolished biofilm formation, and complementation with multicopy wild-type *lapE* (cΔ*lapE*) restored biofilm formation of the Δ*lapE* mutant (Fig. 7B). Dot blot assays using anti-HA antibody revealed that deletion of *lapE* significantly reduced the content of cell surface-associated (CS) LapA, and complementation restored the content of CS LapA. In addition, the c-di-GMP level was elevated in the *lapE* mutant by introducing multiple copies of *wspR*, causing increased total cellular (TC) LapA content, but had no obvious influence on CS LapA content and biofilm formation, indicating that LapE was necessary for LapA secretion and biofilm formation even under high-level c-di-GMP conditions (Fig. 7B). The wild-type strain without the 3× HA tag was used as a negative control (NC), and our data showed no detectable signal in both total cellular LapA and cell surface LapA in this strain. All these results suggested that LapE of *P. putida* was responsible for LapA secretion and biofilm formation.

To further test the role of FleQ and c-di-GMP in biofilm formation and LapA secretion, we constructed an *fleQ* deletion mutant (Δ*fleQ*) and an *fleQ-lapE* double deletion mutant (Δ*fleQ* Δ*lapE*) using the WT-3×HA strain. The *wspR* expression vector was introduced into the two mutants to increase c-di-GMP (Δ*fleQ*+*wspR* and Δ*fleQ* Δ*lapE*+*wspR*), and empty vector was introduced into the mutants as controls (Δ*fleQ*+control and Δ*fleQ* Δ*lapE*+control). The biofilm formation and LapA levels in these mutants were tested. Δ*fleQ*+control showed defection in biofilm formation, as we previously reported (16), and Δ*fleQ*+*wspR* exhibited slightly more biofilm

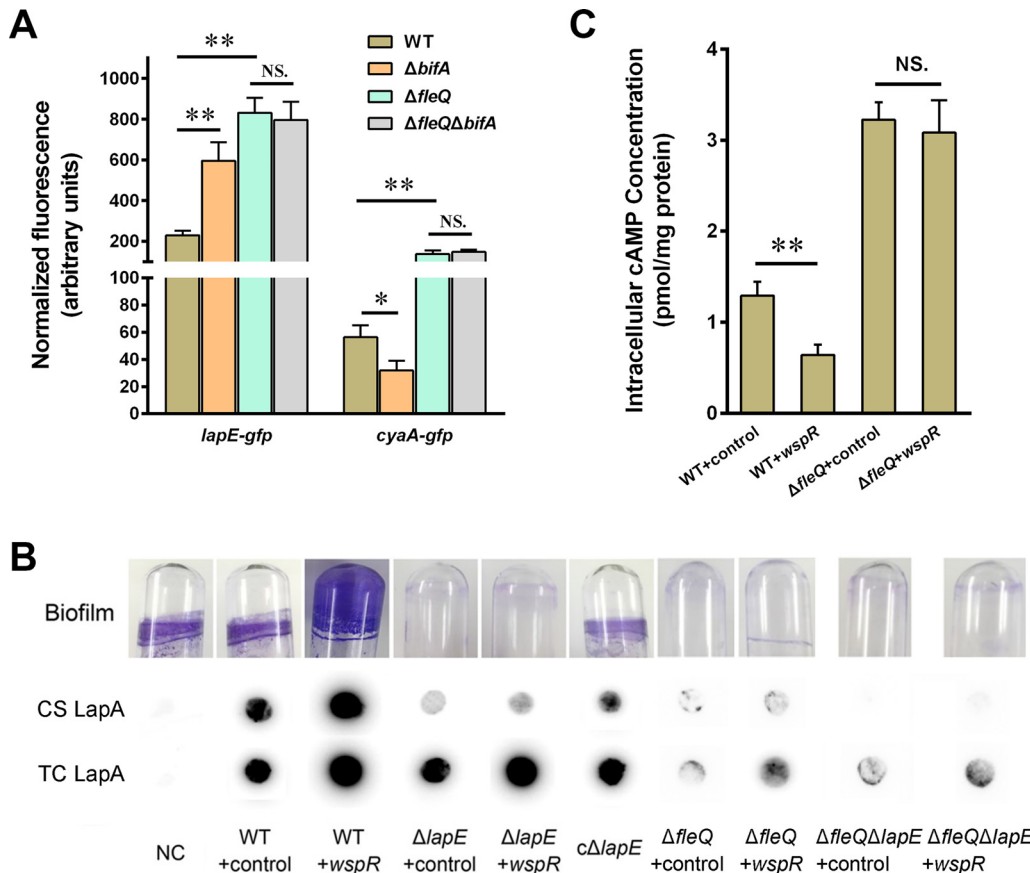

**FIG 7** Functional analysis of c-di-GMP/FleQ-mediated regulation of *lapE* and *cyaA*. (A) Fluorescence intensity of LapE-GFP and CyaA-GFP in the wild type (WT), *bifA* mutant (Δ*bifA*), *fleQ* mutant (Δ*fleQ*), and *fleQ-bifA* double mutant (Δ*fleQ* Δ*bifA*). GFP fluorescence is measured from 24-h growth of LB cultures. (B) Influence of c-di-GMP, *lapE* deletion, and *fleQ* deletion on biofilm formation, cell surface-associated LapA (CS), and total cellular LapA (TC). 3× HA-tagged LapA was tested by using dot blot. The wild-type KT2440 without 3× HA tag was used as a negative control (NC). cΔ*lapE* represents *lapE* mutant complement strains. (C) The cAMP levels in indicated strains. The asterisks above the columns represent statistically significant differences between the two indicated strains (**, $P \leq 0.01$; *, $P \leq 0.05$). NS, not statistically significant between two compared strains. Error bars represent the variant range of the data derived from three biological replicates.

than Δ*fleQ*+control (Fig. 7B). Biofilm formed in Δ*fleQ* Δ*lapE*+control was almost undetectable, and Δ*fleQ* Δ*lapE*+*wspR* showed a similar biofilm phenotype (Fig. 7B). Dot blot assays revealed that Δ*fleQ*+control and Δ*fleQ*+*wspR* exhibited less surface LapA and total LapA than the wild-type strain, which might be attributed to the decreased *lapA* transcription in these two *fleQ* mutants (16). Surface LapA content was undetectable in Δ*fleQ* Δ*lapE*+control and Δ*fleQ* Δ*lapE*+*wspR*, and their total LapA level was significantly lower than that in the wild type but similar to that in Δ*fleQ*+control (Fig. 7B). These results demonstrated that FleQ was essential for c-di-GMP-mediated biofilm formation and LapA secretion.

**c-di-GMP decreases cAMP content in FleQ-dependent manner.** *cyaA*, encoding an adenylate cyclase, is responsible for synthesis of another second messenger, cAMP, in *P. putida* (27). Both a previous study in *P. aeruginosa* and our recent study in *P. putida* revealed that high-level c-di-GMP can decrease the content of cAMP, but their correlation mechanism remains unclear (28, 29). In this study, our result showed that c-di-GMP regulated the expression of *cyaA* in an FleQ-dependent manner. Based on this result, we hypothesized that c-di-GMP decreases cAMP content via FleQ. To test this hypothesis, we measured the cAMP levels in WT+control, WT+*wspR*, Δ*fleQ*+control, and Δ*fleQ*+*wspR* strains by performing an enzyme-linked immunosorbent assay (ELISA). As shown in Fig. 7C, Δ*fleQ*+control exhibited higher cAMP content than

WT+control, which may be attributed to the increased *cyaA* expression in the Δ*fleQ* mutant. WT+*wspR* showed lower cAMP levels than WT+control. However, Δ*fleQ*+*wspR* showed cAMP levels similar to those of Δ*fleQ*+control, indicating that the influence of c-di-GMP on cAMP was abolished in the Δ*fleQ* mutant. This result revealed that c-di-GMP lowered cAMP content in an FleQ-dependent manner.

## DISCUSSION

By performing transcriptomic analysis, we identified 68 differentially expressed operons coregulated by c-di-GMP and FleQ in *P. putida* (Table 1). However, only 50 of them were confirmed to be positive by subsequent qRT-PCR assay (Fig. 3). We speculated that the major reason for this obvious discrepancy was that the three RNA copies used for the transcriptomic assay were extracted from three technical replicates of a single sample, while the RNAs used for the qRT-PCR assay were extracted from three biological replicates of each sample. Therefore, there existed a difference in RNAs between the two assays, and we considered the qRT-PCR result more accurate and reliable in this case. In protein-DNA binding assays (EMSA and B1H), we identified five new target genes under the direct regulation of FleQ. Through ChIP-seq analysis, a previous study identified 160 putative target genes of FleQ in *P. putida* KT2440, including several iron homeostasis-related genes (18). However, these iron homeostasis-related genes have not been identified in this study using transcriptomic analysis. Likewise, *cyaA* and *PP_5586*, identified in this study, have not been identified with ChIP-seq analysis. This result discrepancy may be caused by the different types of applied methods and procedures of transcriptomic analysis versus ChIP-seq analysis. ChIP-seq identifies potential target genes by screening FleQ-bound promoters *in vivo* first and then examining the influence of FleQ on expression of potential target genes, while transcriptomic analysis shows potential target genes by screening differentially expressed genes under the influence of FleQ. Whether FleQ can bind to target promoters is tested in subsequent assays.

FleQ is a global transcriptional regulator, and finding an FleQ-specific binding motif will help to improve target gene identification. Previous studies have reported an FleQ binding motif in *P. aeruginosa* (GTCaNTAAAtTGAC) based on 13 binding sites, and a FleQ binding motif in *P. putida* KT2440 (GTCAaAAAAtTGAC) was identified based on the promoter regions of 15 selected genes and conserved sequence in *P. aeruginosa* (15, 30). In this study, FleQ-binding sites on the promoters of *PP_0681*, *PP_0788*, *lapE*, and *PP_5586* were identified by footprinting assay (Fig. 4C), but no conserved sequence was found from the identified binding sites and no significant similarity was found to the abovementioned two conserved FleQ-binding sequences reported in previous studies. Consistent with this conclusion, the previous ChIP-seq analysis has not found a robust FleQ-binding motif in *P. putida* KT2440 (18). FleQ can form dimers, tetramers, and hexamers, and binding of c-di-GMP to FleQ leads to hexameric ring destabilization and quaternary structure transition disruption, resulting in FleQ existence mainly in the form of monomers and dimers (14, 22). It is possible that oligomerization influences DNA binding preferences of FleQ, which means that FleQ may bind to different sites under different oligomeric states. This hypothesis can explain why no robust FleQ-binding motif has been identified from all the target promoters.

Our data indicated that high-level c-di-GMP and *fleQ* deletion had opposite effects on transcriptions of some target genes (Table 1). For instance, *fleQ* deletion increased transcription of *cyaA* but high-level c-di-GMP decreased transcription of *cyaA* (Fig. 3), and a similar trend was observed in *PP_0788*. The effects of high-level c-di-GMP and *fleQ* deletion on the transcription levels were inconsistent among different genes. For example, for *lapA*, high c-di-GMP caused increased *lapA* transcription, and *fleQ* deletion caused decreased *lapA* transcription (16); for the *bcs* operon, both high-level c-di-GMP and *fleQ* deletion increased *bcs* transcription (16); and for *fleR*, both high-level c-di-GMP and *fleQ* deletion decreased *fleR* transcription (3). One possible explanation is that FleQ functions as both a repressor and an activator to control gene expression by

mSystems®

binding to two sites on the target promoter and that c-di-GMP changes the binding of FleQ to one site, resulting in transcriptional changes under high-level c-di-GMP, as previously described for *pel* regulation in *P. aeruginosa* (13). Consistent with this explanation, our footprinting assay revealed that both *PP_0788*pro and *PP_5586*pro contained two FleQ-binding sites and that both *PP_0681*pro and *lapE*pro had one relatively larger FleQ-binding site (Fig. 4C); this larger binding site might consist of two adjacent binding sites that cannot be distinguished due to experimental limitations.

The adhesin LapA is required for cell surface interactions and biofilm formation in *P. fluorescens* and *P. putida* (21, 31). LapA needs to be retained at the cell surface in a "half-secreted" status to perform its function, and the retention of LapA relies on the outer membrane pore LapE (26). Thus, there might exist a mechanism to ensure matching the expression of LapE and the expression of LapA. This speculation was confirmed by our finding that *lapE* and *lapA* were both positively coregulated by c-di-GMP and FleQ. Promotion of *lapE* transcription by c-di-GMP/FleQ favored LapA secretion and biofilm formation. Under high-level c-di-GMP, LapA was upregulated by the c-di-GMP effector FleQ at the transcription level, while LapA cleavage was inhibited by a c-di-GMP-dependent signaling system at the posttranscriptional level (16, 32–35). Our results in this study demonstrated that LapA was regulated by c-di-GMP at the secretion level. It is these multilevel regulatory mechanisms that guarantee high efficiency of LapA expression/localization and biofilm formation under certain c-di-GMP levels.

Our investigation of another target gene, *cyaA*, reveals the role of c-di-GMP/FleQ in cAMP synthesis regulation, which is in line with the previous findings that high-level c-di-GMP decreased cAMP content in *P. aeruginosa* and *P. putida* (28, 29). However, the mechanism behind this negative correlation between the 2 second messengers remains unknown. Our results showed that transcription of the adenylate cyclase encoded by *cyaA* was directly coinhibited by c-di-GMP and FleQ and that the c-di-GMP-mediated lowering of cAMP content was FleQ dependent, suggesting that c-di-GMP and FleQ coregulate *cyaA* expression and further modulate cAMP in *P. putida*. However, the c-di-GMP-mediated lowering of cAMP content was not caused by inhibition of adenylate cyclase transcription in *P. aeruginosa* (28). The inconsistent results in *P. putida* and *P. aeruginosa* imply that although the same inhibition of cAMP by c-di-GMP was observed in both strains, the inhibition mechanisms are different. After all, functions of cAMP in *P. aeruginosa* and *P. putida* are different. For example, cAMP in *P. aeruginosa* positively regulates the expression of acute virulence factors, including type II and III secretion systems and type IV pili (36, 37), whereas cAMP in *P. putida* is involved in the utilization of amino acids as N sources but plays no roles in virulence regulation (27, 38).

The remaining three target genes, *PP_0681*, *PP_0788*, and *PP_5586*, encode hypothetical proteins with unknown functions. Since c-di-GMP/FleQ is mainly reported to modulate the bacterial plankton-to-biofilm lifestyle transition, the three genes are assumed to belong to unidentified biofilm- or flagellum-related genes, or they may play a role in bacterial plankton-to-biofilm lifestyle transition. Our observation provides evidence for this assumption that *PP_5586* was closely located upstream of the exopolysaccharide Pea synthesizing/transporting operon, implying that *PP_5586* is related to Pea synthesis/transport (39).

In conclusion, we investigated the influence of high-level c-di-GMP and *fleQ* deletion on the transcriptomic profile of *P. putida* and identified five new target genes directly regulated by c-di-GMP/FleQ. We further characterized the function and regulation of two target genes, *lapE* and *cyaA*. The results of this study extend our knowledge of c-di-GMP-mediated FleQ-dependent transcriptional regulation, LapA adhesin secretion, and cAMP synthesis regulation in *P. putida*.

## MATERIALS AND METHODS

**Bacterial strains and growth conditions.** All bacterial strains and plasmids used in this study are listed in Table S5 in the supplemental material. Planktonic cultures of *Escherichia coli* and *P. putida* strains were routinely grown in Luria-Bertani (LB) broth at 37°C and 28°C, respectively, with 180-rpm

shaking. For agar plates, LB medium was solidified with 1.5% (wt/vol) agar. Antibiotics were used, when required, for plasmid maintenance or transformant screening at the following concentrations: kanamycin (50 mg · liter$^{-1}$), chloramphenicol (25 mg · liter$^{-1}$), gentamicin (20 mg · liter$^{-1}$ for *E. coli* or 40 mg · liter$^{-1}$ for *P. putida*), and tetracycline (10 mg · liter$^{-1}$).

**Plasmid and strain construction.** All DNA manipulations were performed by following standard protocols (40). Primers used for plasmid and strain construction are listed in Table S6. All cloning steps involving PCR were verified by commercial sequencing (Tsingke, Wuhan, China). Gene deletion mutants were constructed by following a previously described method (16). Briefly, to construct a *P. putida lapE* deletion mutant, ~800 bp from the chromosomal regions flanking *lapE* were PCR amplified with oligonucleotide pair *lapE_UpS* and *lapE_UpA* (upstream region) or *lapE_DwS* and *lapE_DwA* (downstream region). The PCR products were ligated into a suicide vector, pBBR401, yielding pBBR401-UP-DOWN. A kanamycin resistance cassette was amplified from plasmid pTnmod-RKm and ligated into pBBR401-UP-DOWN, generating the final plasmid pBBR401-UP-Km-DW. The final plasmid was transferred to *P. putida* KT2440 by electroporation. Selection of the kanamycin resistance strain was performed on a kanamycin and chloramphenicol double antibiotics plate. The structure of the deleted *lapE* locus was verified by PCR and sequencing. Other deletion mutants were constructed with the same method.

The strain containing the 3× HA tag-labeled LapA was constructed with a method similar to that for the isogenic mutant, except that a 3× HA epitope-encoding sequence instead of a kanamycin resistance cassette was ligated into the suicide vector containing two adjacent homologous sequences of *lapA*. After electroporation, selection of the integration gentamicin resistance strain was performed on gentamicin and chloramphenicol double antibiotics plates. After subculturing the integration strain in LB medium without antibiotic for six generations, single colonies were obtained by plate streaking. Colonies losing gentamicin resistance were kept for further verification. Finally, 3 × HA-inserted strains were confirmed by PCR and sequencing. The final result was a LapA construct with 3× HA epitopes inserted after residue 7550 in the full-length protein of 8,682 amino acids.

A point mutation of FleQ protein (FleQ$_{K180A}$) was generated by overlapping PCR. Two fragments were generated with primers *fleQ*-K180A1F and *fleQ*-K180A1R as well as *fleQ*-K180A2F and *fleQ*-K180A2R. *fleQ*-K180A1R and *fleQ*-K180A2F were reverse complementary sequences containing the point mutation in which the original AAG codon was replaced by GCG. The two fragments were mixed in a 1:1 ratio, and overlapping extension was performed on a PCR instrument. The final PCR product was digested with EcoRI and BamHI and ligated to pBBR1-MCS5 to yield pBBR1MCS5-*fleQ*$_{K180A}$. The mutation in *fleQ* was confirmed by sequencing. Mutation in the GGDEF motif of WspR (from GGEEF to GGAAF) was generated with the same method.

To construct a *lacZ* reporter plasmid, a fragment (about 500 bp) containing the promoter of the target gene was obtained by PCR. The PCR product was ligated into plasmid pBBR-*lacZ*, which harbored a promoterless *lacZ* gene. The final plasmid was transferred to *P. putida* strains by electroporation, and selection of transformants was performed on a tetracycline and chloramphenicol double antibiotics plate.

**Library preparation, RNA sequencing, and data analysis.** Total RNA from exponentially growing cells in LB medium was extracted with RNA extraction reagent (Vazyme, Nanjing, China) by following the manufacturer's instructions. The RNA samples were depleted of rRNA with a Ribo Zero kit for Gram-negative bacteria (Illumina, USA). cDNA libraries were prepared with a TruSeq stranded mRNA sample preparation kit (Illumina) according to the low-sample LS protocol. Libraries were validated with a DNA 1000 chip on an Agilent 2100 Bioanalyzer, and concentrations were measured using a Qubit 2.0 fluorometer (Invitrogen, Life Technologies). After normalizing each library to 10 nM in TE buffer (10 mM Tris-Cl [pH 7.0], 1 mM EDTA), cDNA libraries were pooled for sequencing on an Illumina HiSeq 2000. The library construction and sequencing were performed at Beijing Novogene Corporation.

RNA-seq data were trimmed using Trimmomatic (41) and analyzed with the open source software Rockhopper (version 2.0.3) with default settings (choosing reverse complement reads and strand-specific analysis) (42). Reads were mapped to the sequenced reference *P. putida* KT2440 genome (GenBank accession no. NC_002947.3). The mapped files were merged using SAMtools, and the identification of novel transcripts was performed by visual inspection with Integrative Genomics Viewer (43, 44), as Rockhopper detected many false positives. The differential gene expression analysis was carried out with the web server T-REx (45) using the reads per kilobase per million (RPKM) values generated in the Rockhopper analysis, in which the *fleQ* mutant or WT+*wspR* was compared to the wild-type control. Differential expression of genes was considered significant with a ≥2-fold change and adjusted *P* value of ≤0.05. The Basic Local Alignment Search Tool (BLAST), with search criteria including a query of >80%, identity of >60%, and E value of <10$^{-6}$, was used in sequence homology searches.

**Quantitative PCR analysis.** Total RNA from exponentially growing cells in LB medium was extracted as described above for transcriptomic analysis. Reverse transcription reactions to generate cDNA were performed with 1 μg RNA using HiScript II Q RT SuperMix (VazymeR223-01). RpoD was used as an internal control for normalization. Primers used for qRT-PCR are listed in Table S6. The degree of change in relative quantity of each target gene was calculated using the $2^{-\Delta\Delta Ct}$ method. Three individual replicates were performed with three independent cultures grown on different days. Standard errors were calculated from these independent replicates.

**Biofilm formation analysis.** Biofilm formation was examined during growth in borosilicate glass tubes without medium replacement as described previously (16). In brief, overnight cultures were 1:100 diluted into fresh LB medium. Incubation was carried out at 28°C for 12 h with shaking. Supernatant was removed, and biomass attached to glass surfaces was visually inspected by crystal violet (0.1%) staining

for 10 min. Excess crystal violet was washed off with distilled water 3 times, and then tubes were left at room temperature for drying for 4 h before taking digital photographs.

**Assays for β-galactosidase activity.** β-Galactosidase activity was measured as described in a previous study (46). Overnight cultures were inoculated (1:100 dilution) in fresh LB medium supplied with tetracycline and grown for 24 h. Two milliliters of culture was pelleted by centrifuging at 4°C, and the pellet was resuspend with the same volume of chilled Z buffer (0.06 mol·liter$^{-1}$ Na$_2$HPO$_4$·7H$_2$O, 0.04 mol·liter$^{-1}$ NaH$_2$PO$_4$·H$_2$O, 0.01 mol·liter$^{-1}$ KCl, 0.001 mol·liter$^{-1}$ MgSO$_4$, 0.05 mol·liter$^{-1}$ β-mercaptoethanol, pH 7.0). After measurement of the optical density at 600 nm (OD$_{600}$), 0.2 ml (V) of the resuspended cell culture was added to 0.8 ml Z buffer, and then 0.1 ml chloroform and 0.05 ml 0.1% sodium dodecyl sulfate were added to permeabilize bacterial cells for 5 min. The β-galactosidase activity reaction was started by adding 0.2 ml of 20 mM o-nitrophenyl-β-galactopyranoside (in Z buffer). After the reaction solution turned yellowish at 28°C, 0.5 ml of 1 M Na$_2$CO$_3$ was added to stop the reaction, and reaction time (T), in minutes, was record. The reaction time was limited to within 30 min, and if the β-galactosidase activity was too low to turn the reaction solution yellowish within 30 min (such as cyaApro-lacZ), more cell culture (0.5 ml) would be used in the reaction mixture. Finally, we spun the reaction solution for 5 min at 15,000 × g to remove debris and chloroform, and the optical densities of the supernatant at 420 nm and at 550 nm were measured with a spectrophotometer (INESA, Shanghai, China). The unit of enzyme activity was calculated using the following equation: Miller units = 1,000 × [(OD$_{420}$ − 1.75 × OD$_{550}$)]/(T × V × OD$_{600}$). The wild-type strain with multiple wspR genes was repeatedly blown with a pipette until the culture became well distributed before analysis, since the strain formed clumps and strong biofilm. Measurements were repeated at least in triplicate with two technical repeats per sample.

**Fluorescence measurements.** Overnight cultures bearing reporter plasmid lapE-gfp or cyaA-gfp or pCdrA::gfp$^C$-tet (a derivate of pCdrA::gfp$^C$, with its gentamicin resistance gene replaced by a tetracycline resistance gene) were diluted in 5 ml LB to an OD$_{600}$ of 0.01. Cultures were incubated for 24 h at 28°C with shaking. For GFP fluorescence measurements, culture samples were transferred to 96-well microtiter dishes, and OD$_{600}$ and fluorescence were determined on an Envision mutimode plate reader (PerkinElmer, Germany). Fluorescence was measured from 150-μl samples using a 485-nm band pass filter and a 520-nm emission filter on a black 96-well plate (Costar, USA). Specific fluorescence was calculated by dividing the fluorescence reading by the OD$_{600}$ reading and then normalized by subtracting the autofluorescence of wild-type or bifA mutant culture not bearing a GFP plasmid. The WT+wspR and bifA mutant cultures were repeatedly blown with a pipette until the culture became well distributed before analysis, since the strain formed clumps and strong biofilm. For each measurement, 4 biological replicates were assayed in sextuplate.

**Dot blot assay and Western blot assay.** Dot blot assay was performed as described in a previous study (47). Briefly, aliquots of exponentially growing cultures were pelleted, washed once with phosphate-buffered saline (PBS), and then resuspended in PBS. For cell surface-associated LapA assays, cell suspensions were adjusted to equivalent OD values (OD$_{600}$ ≈ 1), and then 5-μl aliquots of each suspension were spotted onto a nitrocellulose membrane. For total cellular LapA assays, cells were lysed with a JNBIO pressure cell breaking apparatus, and protein concentration was determined by bicinchoninic acid (BCA) assay. The same amount of protein (5 μg) was spotted onto a nitrocellulose membrane. After drying at room temperature, membranes were probed for LapA-3×HA using an anti-HA mouse antibody (Signalway Antibody). A horseradish peroxidase-conjugated secondary antibody (goat anti-mouse; Signalway Antibody) was used for chemiluminescent detection of bound ligands. Detection was carried out using Western ECL reagents (Bio-Rad). Images of dot blots were digitized using a Tanon 5200 scanner (Shanghai, China).

For Western blot assay, strains harboring strep II-tagged WspR were cultured and lysed with the method described above for the dot blot assay. The same amount of protein from different samples was resolved by 12.5% SDS-PAGE. After transferring the proteins onto a nitrocellulose membrane, WspR proteins on the membrane were tested with anti-strep II tag mouse antibody (Sangon Biotech, China). A horseradish peroxidase-conjugated secondary antibody (goat anti-mouse; Signalway Antibody) was used for chemiluminescent detection of bound ligands.

**Expression and purification of His-tagged FleQ.** E. coli BL21 carrying pET28a-fleQ was grown overnight in LB, diluted 1:100, and grown for 4 h at 37°C. The expression of His-tagged FleQ was induced with addition of 0.4 mM IPTG (isopropyl-D-thiogalactopyranoside), followed by incubation at 16°C overnight. Cultures were harvested by centrifugation at 6,000 × g for 10 min and resuspended in lysing buffer (10 mM Tris-Cl [pH 7.8], 300 mM KCl, and 10% glycerol). Cells were lysed with a JNBIO pressure cell breaking apparatus, followed by centrifugation at 15,000 × g for 5 min. The lysate was filtered through a 0.22-μm-pore-size filter before it was loaded onto a NiSO$_4$ column and collected from the column with elution buffer (10 mM Tris-Cl [pH 7.8], 300 mM KCl, 10% glycerol, 250 mM imidazole). Protein concentration was determined by BCA assay.

**EMSA.** Fragments of target promoters used in EMSA were generated by PCR using 6-FAM (6-carboxyfluorescein phosphoramidate)-labeled primers (Tsingke, Wuhan, China). Equal amounts of labeled DNA fragments (0.16 pM) were added to binding reactions with various amounts of FleQ in binding buffer (10 mM Tris, pH 7.8, 10 mM magnesium acetate, 50 mM KCl, 5% glycerol, 250 ng ml$^{-1}$ bovine serum albumin, 20 μl total reaction volume). FleQ was incubated with DNA for 30 min at 25°C. Reaction mixtures containing c-di-GMP were performed as described above, except that c-di-GMP was incubated with FleQ for 10 min before addition of DNA. All reaction solutions were loaded onto a 5% acrylamide gel containing 10 mM Tris-Cl (pH 7.8), 400 mM glycine, 5 mM EDTA and electrophoresed at 100 V at room temperature for 1 to 1.5 h. Gels were dried and exposed to a phosphorimaging screen.

**Bacterial one-hybrid assay.** FleQ was cloned into pTRG vector. Promoters of target genes were amplified and cloned into pBXcmT vector (21). Primers used for amplifying target promoters are listed in Table S6. *E. coli* XL1-Blue MRF' Kan (Stratagene) was used for routine propagation of all pBXcmT and pTRG recombinant plasmids. Bacterial one-hybrid assays were carried out as described previously (21). Positive growth cotransformants were selected on a selective screening medium plate containing 10 mM 3-AT, 10 $\mu$g/ml streptomycin, 10 $\mu$g/ml tetracycline, 25 $\mu$g/ml chloramphenicol, and 30 $\mu$g/ml kanamycin. The plates were incubated at 30°C for 3 days. Cotransformants containing the pBXRv2031/ pTRG-Rv3133 plasmids (21) served as a positive control, and cotransformants containing empty vectors pBXcmT and pTRG served as negative controls.

**DNase I footprinting assay.** The 6-FAM-labeled target promoters were PCR amplified using 6-FAM-labeled primers. In a 600-$\mu$l reaction system, 1,000 ng of labeled DNA fragment was bound to 600 nM FleQ (final concentration; bovine serum albumin was used instead of FleQ in the control experiment) in buffer containing 10 mM Tris-HCl (pH 7.8), 10 mM $MgCl_2$, 1 mM $CaCl_2$, 0.4 mM dithiothreitol, 100 mM KCl, and 5% glycerol and incubated for 30 min at room temperature. After binding, 0.03 U of RNase-free DNase I (Roche, Basel, Switzerland) was added and allowed to react for 3 min at 25°C. The reaction was stopped and precipitated with ethanol. Samples were analyzed in a 3730 DNA Analyzer (Applied Biosystems, Foster City, CA, USA), and the electropherograms were aligned with GeneMapper v3.5 (Applied Biosystems).

**cAMP content measurement.** Intracellular cAMP contents of *P. putida* were measured using an enzyme-linked immunosorbent assay (ELISA) kit (Abcam). Cells were grown in M9 medium supplemented with 0.4% glucose as a carbon source for 24 h. Nucleotides were extracted with extraction solution (methanol-acetonitrile-water, 2:2:1, vol/vol/vol) and a heating method described previously (28). The samples were assayed for cAMP by following the manufacturer's protocol. The protein concentrations of bacterial samples were measured by the BCA protein assay. cAMP contents were normalized to the total protein per milliliter of culture.

**Statistical analysis.** For analysis of the significance of differences in gene expression and cAMP concentrations, Student's *t* test was used for comparison of two groups of data. For analysis of the significance of differences in $\beta$-galactosidase activity, analysis of variance was used for comparison of three or more groups of data. A *P* value of less than or equal to 0.05 was considered statistically significant.

**Data availability.** Transcriptomic data were deposited in the SRA database under accession numbers SRR8959868, SRR8959867, SRR8959870, SRR8959869, SRR8959872, SRR8959871, SRR8959874, SRR8959873, and SRR8959875.

## SUPPLEMENTAL MATERIAL

Supplemental material is available online only.
**FIG S1**, TIF file, 1.9 MB.
**FIG S2**, TIF file, 0.3 MB.
**FIG S3**, TIF file, 2.2 MB.
**TABLE S1**, DOC file, 0.3 MB.
**TABLE S2**, DOC file, 0.2 MB.
**TABLE S3**, DOC file, 0.3 MB.
**TABLE S4**, DOC file, 0.3 MB.
**TABLE S5**, DOC file, 0.1 MB.
**TABLE S6**, DOC file, 0.2 MB.

## ACKNOWLEDGMENTS

The research was financially supported by the National Key Research and Development Program of China (2018YFE0105600), National Natural Science Foundation of China (41830756 and 31900054), and the China Postdoctoral Science Foundation (2018M642861).

We thank Weihui Li from Guangxi University (Nanning, China) for helping us revise the manuscript and providing valuable suggestions for our study. We also thank Ping Liu from Huazhong Agriculture University (Wuhan, China) for her work on English editing and language polishing.

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
