## [Reviewer comments · mSystems]

Identification of c-di-GMP/FleQ-regulated new target genes including *cyaA* encoding adenylate cyclase in *Pseudomonas putida*

Yujie Xiao, Haozhe Chen, Liang Nie, Meina He, Qi Peng, Wenjing Zhu, Hailing Nie, Wen-Li Chen, and Qiaoyun Huang

Corresponding Author(s): Wen-Li Chen, Huazhong Agricultural University

Review Timeline:

Submission Date:

March 11, 2021

Accepted:

April 10, 2021

Editor: Jonathan Eisen

Reviewer(s): The reviewers have opted to remain anonymous.

Transaction Report:

DOI: <https://doi.org/10.1128/mSystems.00295-21>

Response to reviewer 1:

Reviewer #1 (Comments for the Author):

This revised manuscript describes experiments that explore new targets of FleQ/cdG, including the AC in *P. putida*. They use transcriptomics and DNA binding assays to explore these questions, and report that the 5 new genes were identified as part of the FleQ/cdG regulation: PP_0681, PP_0788, PP_4519 (*lapE*), PP_5222 (*cyaA*), and PP_5586. They conclude that FleQ/cdG inhibits transcription of PP_0788 and *cyaA*, and promotes transcription of PP_0681, *lapE*, and PP_5586. They present evidence that FleQ/cdG regulation is likely direct via DNA binding to the promoter by this transcription factor. They propose that FleQ/cdG regulation of *LapE* impacts *LapA* cell surface localization and biofilm formation, and that FleQ/cdG controls production of cAMP, linking the regulation of these nucleotide effectors, which has been observed previously.

The strengths of this manuscript include some interesting observations, including linking a new set of likely direct targets to the FleQ/cdG regulon in *P. putida*, and providing a possible mechanism for the previous observations that cAMP and cdG are inversely regulated. These are observations that will be useful in the field. The weakness of the manuscript includes the writing, which includes some tortured text and phrasing that is often somewhat misleading (see comment #1) below, which will make what appear to be interesting data somewhat challenging to access by the reader. The manuscript needs SIGNIFICANT editing. There are also a number of experimental shortcomings highlighted below - perhaps less would be more here, and just focus on the key findings.

Our response:

Thanks a lot for all the comments and suggestions. We learned a lot from them. We have revised the manuscript according to these comments and suggestions. English writing is one of our weaknesses. To improve our writing, we have invited a linguistics professor to help us editing and polishing language of our manuscript.

Specific comments:

1. Overall, the initial transcriptome analyses conducted in these experiments are reasonable, however, additional transcriptome analyses comparing a $\Delta fleQ$ and $\Delta fleQ+WspR$ strains to identify genes that are cdG-regulated but are not FleQ regulated. This list would need to be compared to the 133 genes originally identified in the original screen.

Our response:

Thanks a lot for this inspiring suggestion, but the main topic of this study is genes

regulated by c-di-GMP via FleQ, thus, identifying genes that are c-di-GMP-regulated but are not FleQ regulated is not concerned much here. However, this is a useful suggestion, for we are planning to investigate other function of c-di-GMP, except modulating biofilm formation and motility. Finding genes that are c-di-GMP-regulated but are not FleQ regulated will surely give us some inspirations. Besides, generally, transcriptional regulation mediated by c-di-GMP requires transcriptional regulatory effectors like FleQ. So far, FleQ is the only identified transcriptional regulatory effector in *P. putida*, but we believe that there are other transcriptional regulatory effectors in this strain, and we are currently performing screening experiment to find potential new transcriptional regulatory effectors. Identifying genes that are c-di-GMP-regulated but not FleQ-regulated in advance may also help to find new transcriptional regulatory effector.

2. Line 22/Line 33 Not all 5 of these genes are new targets. Some of these genes were identified in Blanco-Romero et al. 2018 Scientific Reports.

Our response:

Yes, PP_0681, PP_0788, and PP_4519 were identified as target of FleQ in a ChIP-Seq analysis by Blanco-Romero et al. in 2018, but competition assays were not performed to confirm whether the binding of FleQ to their promoters was specific or not. Besides, function of c-di-GMP in regulating these target genes was not been studied in their study. Our results confirmed the specific binding of FleQ to promoters of these target genes, and function of c-di-GMP was also investigated, thus, we termed these genes as “new target genes regulated by c-di-GMP/FleQ” in our study.

3. Lines 43-45. "Regulation of lapE by c-di-GMP/FleQ is a new strategy of the bacteria to guarantee high efficiency of LapA expression and biofilm formation under certain c-di-GMP level". Careful here - I think you mean that control of lapE expression could impact LapA localization to the cell surface - using the phrase "LapA expression" here implies that FleQ/cdG is controlling expression of the gene encoding LapA, which is not what I think you are trying to say.

Our response:

Thanks a lot for this comment. We have revised the sentence as “Regulation of lapE by c-di-GMP/FleQ guarantees high efficiency of LapA localization and biofilm formation.” in the new manuscript. The revision is shown in page 3 lines 47-49 in the marked-up manuscript.

4. Lines 61-64. It is not clear, as written, that all the transcription factors you are

describing here are cdG-responsive. Pls clarify by modifying the text.

Our response:

We have modified the text to make it clear that all the transcriptional regulators mentioned are c-di-GMP-responsive. The revision is shown in page 4 lines 65-74 in the marked-up manuscript.

5. Line 76-78. Rework that sentence - you are missing at least on word.

Our response:

The sentence has been revised as "FleQ functions as both a repressor and an activator to bind to two sites on the promoter of exopolysaccharide pel operon, and it controls the activity of pel promoter along with FleN (another ATPase) in response to c-di-GMP in *P. aeruginosa*." The revision is shown in page 5 lines 88-91 in the marked-up manuscript.

6. Line 79: "with the two FleQs" - this is jargon - rework.

Our response:

The "two FleQs" has been changed to "two FleQ molecules". The revision is shown in page 5 lines 91-92 in the marked-up manuscript.

7. Line 88-89. Rework this sentence "the knowledge on FleQ" to "the knowledge of FleQ"

Our response:

The sentence has been reworked by changing "the knowledge on FleQ" to "the knowledge of FleQ". The revision is shown in page 6 line 103 in the marked-up manuscript.

8. Line 89: should be "sequences"

Our response:

The "sequence" has been changed to "sequences". The revision is shown in page 6 line 103 in the marked-up manuscript.

9. Line 91: "which is important" → "which are important"

Our response:

The sentence has been changed to "...such as *siaABCD* operon and *bdIA* gene in *P. aeruginosa* respectively responsible for cell aggregation and biofilm dispersal,...". The revision is shown in page 6 lines 105-107 in the marked-up manuscript.

10. Line 92: "homologue to *gcbA*," → "homologue of *gcbA*,"

Our response:

The "homologue to gcbA," has been changed to "homologue of gcbA,". The revision is shown in page 6 line 107 in the marked-up manuscript.

11. Line 98" "some common part of its direct regulon" - rework this sentence

Our response:

The sentence has been revised as "FleQ shares some common target genes with another global regulator AmrZ in *P. fluorescens*,". The revision is shown in page 6 lines 112-114 in the marked-up manuscript.

12. Note multiple edits above the clarify the text in the intro. As I am not a copy editor, and there are issues like this every couple of lines, I cannot do this editing for the whole manuscript - I would suggest a professional copy editor to help with the writing.

Our response:

Thanks a lot for these edits points. They have improved our manuscript largely, and we have learned a lot from these edits. To improve our writing, we have invited a linguistics professor to help us editing and polishing language of our manuscript.

13. Line 114: "three-fold increase in WT+wspR relative to WT+control" do you mean a 3-fold increase in cdG between the two strains?

Our response:

No, it is a three-fold increase in GFP fluorescence in WT+wspR relative to WT+control. Higher GFP fluorescence represents higher intracellular c-di-GMP concentration, but we cannot say that it's three-fold increase in c-di-GMP. We have revised the sentence as "Normalized fluorescence results revealed an about three-fold increase in GFP fluorescence of WT+wspR relative to that of WT+control, indicating that introducing pBBR1MCS5-wspR to wild-type provokes an increase in cellular c-di-GMP.". The revision is shown in page 7 lines 132-135 in the marked-up manuscript.

14. Line 116-121. I cannot find anywhere how many replicates were performed for the RNA-Seq experiment. Please state this information explicitly in the text - one should not have to search for this information. A similar issue is noted for the section starting on line 129.

Our response:

Thanks a lot for this comment. Three technical replicates from one biological replicate were performed in the RNA-seq experiment, which means that the RNA

copies used for the RNA-seq were extracted from three replicates of one sample. This limitation had been discussed in the first paragraph of discussion section. We have stated this information explicitly in the result section of the new manuscript. The revision is shown in page 7 lines 138-139 and page 8 line 156 in the marked-up manuscript.

15. Line 159-166. It is not clear from this section if the qPCR studies were done under the same conditions as the RNA-Seq studies (cells grown for the same times, same medium, etc). Any thoughts on why 18 of the 68 genes did not replicate? Seems high as transcriptomic data often under-represents changes. Could this be due to difference growth conditions - pls clarify.

Our response:

Thanks a lot for this comment. The qPCR studies were done under the same conditions as the RNA-Seq studies. RNAs used for qPCR and RNA-Seq were extracted from exponentially growing cells (12 hours incubation at 28°C, with 180 rpm shaking) in LB medium following same protocol. We infer that the most important reason for the obvious discrepancy between transcriptomic and qRT-PCR results is that the three RNA copies used for the transcriptomic assay were extracted from three technical replicates of one single sample, while the RNAs used for the qRT-PCR assay were extracted from three biological replicates of three individual samples; therefore, some discrepancy between RNAs for the two techniques may exist, and we consider the qRT-PCR result more precise under this condition. After we had checked the result of transcriptomics with qPCR and found that 18 of the 68 genes did not replicate, we did realize that using 3 technical replicates from 1 biological replicate in RNA-Seq was not reliable. Although the main target genes found in RNA-Seq were checked by using qRT-PCR and promoter-lacZ fusion reporter, which could make up for this limitation to some extent, but still that was a weak experimental design, and concerns about this weak design had also been mentioned by previous reviewers. We will remember the lesson and be more careful in designing our future studies.

16. There appear to be several band shifts in Fig. S1 (i.e., PP_4519 and PP_5586) - why were they not included in the analysis? They do seem to drive a shift - please explain.

Our response:

Fig. S1 shows all EMSA results of the 50 target promoters, including the five main target promoters. The PP_4519 (also named lapE) and PP_5586 are already included in the analysis (Fig. 4, Fig. 5, Fig. 6).

17. The one-hybrid assay does seem to nicely confirm the EMSA for the 5 genes focused on in this proposal. Were the other proteins I mentioned above (i.e., PP_4519 and PP_5586) tested in this assay?

Our response:

As explained for comments 16. The PP_4519 (also named lapE) and PP_5586 are already included in the analysis.

18. Much more FleQ protein was used in the DNases footprinting than the EMSA. Any reason why?

Our response:

This is based on the characteristics of DNase I footprinting assay. To find the protecting region on DNA, one should try to guarantee that every DNA molecular in the mixture is bound by protein. In the EMSA assay, we found that FleQ at 300 nM can almost cause shift of all DNA molecules on the gel, thus to obtain a fully binding status, we decided to use 600 nM FleQ in the DNase I footprinting assay.

19. I am concerned about the *cya* direct binding conclusion. The shifts are occurring at high concentrations, and no footprint is obvious, though the one hybrid is consistent with binding. I think you need to deal with this gene distinctly in the abstract and discussion as a POSSIBLE direct target, with some ambiguous data.

Our response:

This was also our concern while performing the binding assay, the binding of FleQ to *cyaA* promoter was weaker than that of the other four promoters. We performed the binding assay for three times with newly purified protein, and the results were similar, indicating that it is indeed a weak binding. Besides, we failed to confirm the precise binding site on *cyaA* promoter using DNase I footprinting. However, the bacterial one hybrid assay showed positive binding result. To map the specific region of the *cyaA* promoter interacting with FleQ, we truncated and divided the *cyaA* promoter into three fragments, and then performed EMSA with these fragments. The results showed that two fragments (*cyaApF2* and *cyaApF3*) produced band shifts with FleQ on the gel, whereas no band shift was observed with *cyaApF1* (Fig. 4D), indicating that the binding site locates between position -139 and -51 on the *cyaA* promoter relative to its translational start site. Besides, the band shifts became stronger when the concentration of FleQ increased (Fig. 4D), and the band shifts was similar to other target genes when FleQ concentration reached 400 nM. Thus, we infer that *cyaA* is a direct target of FleQ, but the affinity of FleQ to *cyaA* is weaker than other promoters, and this may explain that

influence of FleQ on expression of *cyaA* is weaker than other target genes. Together, our results demonstrate that FleQ binds to *cyaA* promoter directly, though it is a weak binding compared with other promoters.

20. The expression data in Figure 5 indicates two things. First, for 4/5 genes it appears that FleQ represses expression, and in the absence of FleQ, no additional upregulation is observed when cdG is modified. This is the case even for the one promoter (PP_0078) where the *wspR* expression reduced expression modestly - loss of FleQ still resulted in a 2-3-fold increase in expression of PP_0078. Second, I would argue, at best cdG and FleQ effects are very modest of *cyaA* - yes the changes are significant, but are these small changes biologically significant?

Our response:

FleQ represses expression of all the five genes, since that deletion of FleQ led to increased activity of all five promoters. However, c-di-GMP+FleQ showed different influence on activity of these promoters. Based on previous studies, there seems no regularity between the transcription outcomes of high c-di-GMP levels and fleQ deletion among different target genes. For example, in transcription of *lapA*, high c-di-GMP caused increased *lapA* transcription, and fleQ deletion caused decreased *lapA* transcription; in transcription of *pel* operon, both high c-di-GMP and fleQ deletion caused increased *pel* transcription; in transcription of *fliF*, a flagellar-related gene, both high c-di-GMP and fleQ deletion caused decreased *fliF* transcription; in transcription of PP_0788 and *cyaA*, two target genes identified in this study, high c-di-GMP caused decreased PP_0788 and *cyaA* transcription, and fleQ deletion caused increased PP_0788 and *cyaA* transcription. One model for *pel* regulation in *P. aeruginosa* explains how c-di-GMP/FleQ regulates *pel*. FleQ binds to two sites on *pel* promoter, one for activation and another for repression. c-di-GMP could change the binding of FleQ to one site, resulting transcriptional changes under high and low c-di-GMP levels. Consistent with this model, our footprinting assay revealed that both PP_0788pro and PP_5586pro contained two binding sites of FleQ, and both PP_0681pro and *lapE*pro had one larger binding site (Fig. 4C), which may consist of two adjacent binding sites that can not be distinguished due to experimental limitation.

Promoter activity of *cyaA* in WT+wspR and fleQ mutant had been measured for two times, with three biological replicates each time, and the trend of two results was similar. Thus the effects of c-di-GMP and FleQ on *cyaA* are biologically significant. Besides, the cAMP concentration in WT+wspR is significantly lower than that in WT, and this trend was also observed in three biological replicates. The reason of modest regulation on *cyaA* may relate to the weak binding of fleQ to

cyaA promoter as shown in the EMSA assay.

21. Do you have any evidence that the GGEEF to GGAAF mutant of WspR is stable? I did not see these data. These data are important to make this conclusion: "...implying that the transcription changes of the five genes was caused by increased c-di-GMP levels, not unintended effects caused by overexpression of WspR."

Our response:

Thanks a lot, this comment is very enlightening to us. We had taken it for granted that the GGEEF to GGAAF mutant of WspR would abolish its DGC activity, so it can not raise the c-di-GMP levels, but we had not considered whether the point mutation would influence stability of WspR. This question is important for our conclusion.

To remedy this deficiency, we have recently cloned the mutated wspR to an expressional vector, and a Strep II tag was fused to the N terminal of WspR, so we can detect the WspR protein using western-blot. An expressional vector with Strep II tag fused to wild-type wspR was also involved as positive control. Then the vectors were introduced to wild-type KT2440 and fleQ deletion mutant. The WspR proteins in the two strains were detected using western-blot with anti-strep II antibody. The results turned out that the point mutated WspR could be detected in both wild-type strain and fleQ deletion mutant, with molecular weight and signal intensity similar to that of the wild-type WspR protein, indicating that the point mutated WspR is stable in both wild-type and fleQ deletion mutant. These results has been added to the new manuscript as new Fig S2, and related description was added to the results part. The revision is shown in page 15 lines 296-300 in the marked-up manuscript.

22. Line 241-244: have these FleQ mutants been shown to be stable? This is less of an issue given the lack of phenotype, but this point should be mentioned.

Our response:

These FleQ mutants had been constructed in *P. aeruginosa* before, and their protein structures were also analyzed (Matsuyama et al., 2016. Proc Natl Acad Sci USA 113:E209–E218; Baraquet and Harwood, 2013. Proc Natl Acad Sci USA 110:18478–18483). Alignment result showed that these amino acid residues were the same in FleQ from *P. aeruginosa* and FleQ from *P. putida*, thus we consider that these FleQ mutants are stable.

23. The lack of impact of the FleQK180A on cyaA expression also suggests that the mechanism whereby FleQ modestly impacts cya is distinct from the other genes. This

point needs to be made clear.

Our response:

Thanks a lot for this comment. The FleQ_{K180A} can repress activities of all five target promoters (Fig. 6), since that the activities of all five promoters decreased in cΔfleQ_{K180A} compared with that in fleQ mutant, which means that the FleQ_{K180A} can complement the fleQ mutant as wild-type FleQ does. But, we also found that the FleQ_{K180A} showed better repression ability than wild-type FleQ for PP_0681pro, PP_0788pro, lapEpro and PP_5586pro, since promoter activity of the four promoters in cΔfleQ_{K180A} was lower than that in cΔfleQ_{WT}. But promoter activity of cyaA was not influenced by this point mutation, the FleQ_{WT} and FleQ_{K180A} showed similar repression ability on activity of cyaApro (Fig. 6). This is an interesting result, but we can't explain it at the moment. We have added description about this result in the result part in the new manuscript. The revision is shown in page 18 lines 357-359 in the marked-up manuscript.

24. As mentioned by the authors, the presence or absence of cdG can change the oligomerization of FleQ and thus change the ability of FleQ to bind to a promoter and change the location of FleQ/promoter binding. The DNase I Footprinting assays presented in the study do not utilize cdG in the protocol and thus provide an incomplete picture as to where FleQ is binding the five promoters, and particularly the one instance where footprinting could not be demonstrated. Have you tried varying concentrations of cdG in order to identify any alternative binding site of FleQ?

Our response:

Thanks a lot for this comment and suggestion. We have recently re-performed DNase I footprinting assay with c-di-GMP added for all five promoters. However, we found no obvious difference for the binding sites with or without c-di-GMP in the reaction mixture. This may cause by the low sensitivity of footprinting assay. To test whether and how c-di-GMP affects binding of FleQ to the target promoters, we added c-di-GMP in the EMSA assays, in which the c-di-GMP was added at a concentration from 0 to 90 μM. The results showed that c-di-GMP enhanced binding of FleQ to PP_0681pro, lapEpro, and 5586pro, but showed no obvious influence on binding to PP_0788pro and cyaApro. We have added these results to the new manuscript. The revision is shown in pages 13-14 lines 254-270 in the marked-up manuscript.

25. Line 262-263: "These results demonstrated that the c-di-GMP binding ability of FleQ was not required for complementation of PP_0681, PP_0788, lapE, cyaA and

PP_5586". How do you square this observation with the data above suggesting that FleQ and cdG appear to regulate these genes? This is somewhat confusing...

Our response:

This is caused by the very nature of FleQ. FleQ is a transcriptional regulator, and it can bind to target promoter without c-di-GMP. The binding causes activation/repression of target genes. C-di-GMP functions as a switch of FleQ, which can change its activity by binding to it. So without c-di-GMP, FleQ alone can still bind these target promoters, and activates/represses the promoters. Here we used the FleQ_{K180A}, which has no c-di-GMP binding ability, and it can still complement the fleQ mutant by repressing the five promoters.

26. The idea of the FleQ domain analysis in Figure 6 is good in theory, however, there are some issues with the design of this experiment. The point mutations identified in Baraquet and Harwood, 2013 were identified in *P. aeruginosa* rather than *P. putida* so you cannot assume that the position of all the amino acid residues in FleQ is the same between the two species. Have you done alignments to confirm the likely overlapping role of these residues?

Our response:

Thanks a lot for this comment. We had done the alignment work before we constructed those point mutated FleQ, but the result was not shown. FleQ is highly conserved regulator among *Pseudomonas* species. The FleQ from *P. aeruginosa* and FleQ from *P. putida* shares 84% amino acid sequence identity with each other, and the three amino acid related to point mutation is the same between the two species. We have added the alignment result to the new manuscript as Fig. S3, and related description was added to the results part. The revision is shown in page 16 lines 323-327 in the marked-up manuscript.

27. Figure 7. Do you have any indication as to whether these GFP-fusions are active?

Our response:

No. The two GFP-fusions were constructed to check whether the transcriptional changes of *lapE* and *cyaA* lead to associated protein level changes, and transcription of the GFP-fusions on the plasmid was controlled by *lapE/cyaA* promoter. The two GFP-fusions can emit fluorescence when excited with 485 nm light, indicating that the GFP part is active. The fluorescence intensity represents GFP protein level, which equals to *LapE/CyaA* protein level.

28. The experiment in Figure 7 is interesting but I do not believe that you establish that the increase in secretion of *LapA* when *cdG* is high is due specifically to an

increase in LapE level. You show LapA on the cell surface is indeed LapE-dependent (consistent with previous work) and that increased cdG increases LapA levels overall and on a surface, but the dependency of increased LapE is not established here. For example, could there be more LapA on the surface because there is more made??

Our response:

Thanks a lot for this comment. Your concern is right, we shouldn't use that title and claim that "C-di-GMP/FleQ modulates LapA secretion via regulating expression of lapE." Both expression of lapA and lapE was positively regulated by c-di-GMP, and our result here showed that LapA on the cell surface was LapE-dependent. When c-di-GMP was high, both lapA and lapE increased. There is a possibility that secretion of lapA is modulated by the level of lapE, which is pretty reasonable, but we have no direct evidence to support that conclusion. We have revised the title of that paragraph as "LapE is responsible for LapA secretion and biofilm formation". The revision is shown in page 20 line 406 in the marked-up manuscript.

29. For Figure 7A, a WT and Δ bifA strain only tells you the effect of cdG on LapE and CyaA protein expression but does not tell you anything about FleQ-dependent regulation. Here, you would need to include Δ fleQ and Δ fleQ Δ bifA strains.

Our response:

Thanks a lot for this comment. We have recently constructed a Δ fleQ Δ bifA strain, and introduced the two gfp-fusion vectors into Δ fleQ and Δ fleQ Δ bifA. Together with the two formerly constructed strains (WT and Δ bifA), we measured fluorescence intensity of GFP in these strains. The results showed that GFP fluorescence intensity of the bifA mutant (Δ fleQ) containing lapE-gfp was much stronger than that of wild-type, while the fluorescence intensity of cyaA-gfp showed an opposite trend (Fig. 7A). Fluorescence intensity of both fusion proteins in fleQ mutant is stronger than that of wild-type, but deletion of bifA in the fleQ mutant showed no obvious influence on fluorescence intensity, suggesting that the modulation of c-di-GMP on lapE and cyaA protein levels is FleQ dependent. These results indicated that transcription changes of lapE and cyaA under high c-di-GMP levels led to changes in protein level in a FleQ dependent manner. We have added these results to the new manuscript. The revision is shown in pages 19-20 lines 387-405 in the marked-up manuscript.

30. The small decrease in cAMP is indeed consistent with the model you present about hi cdG leading to low cAMP. But again, not all the data line up and this should be made clear to the readers.

Our response:

As mentioned above in the response to the 20th comment, three biological replicates were used in the assay of *cyaA* expression of cAMP concentration, the measurements were performed twice, and similar results were obtained. Thus the effects of c-di-GMP/FleQ on *cyaA* expression and cAMP level are biologically significant. The reason of modest regulation on *cyaA* and cAMP may relate to the weak binding of fleQ to *cyaA* promoter as shown in the EMSA assay.

31. For the biofilm analyses done in Figure 7B, you should also include a Δ fleQ + control and Δ fleQ +WspR strains as well as a Δ fleQ Δ lapE + control and Δ fleQ Δ lapE +WspR strains. This would demonstrate that these phenotypes are FleQ and cyclic-di-GMP dependent.

Our response:

Thanks a lot for this comment. We have constructed a Δ fleQ and a Δ fleQ Δ lapE using wild-type/ Δ lapE containing 3 \times HA tag lately. The *wspR* expressional vector and control vector were introduced to the mutants to obtain Δ fleQ+control, Δ fleQ+wspR, Δ fleQ Δ lapE+control, and Δ fleQ Δ lapE+wspR. Then biofilm formation and *lapA* levels were tested. Δ fleQ+control showed defection in biofilm formation, and Δ fleQ+wspR exhibited slightly more biofilm than Δ fleQ+control. Biofilm formed in Δ fleQ Δ lapE+control was almost undetectable, and Δ fleQ Δ lapE+wspR showed similar biofilm phenotype. Dot blot assays revealed that Δ fleQ+control and Δ fleQ+wspR exhibited less amount of cell surface LapA and total LapA than wild-type strain, which might be attributed to the decreased *lapA* transcription in these two fleQ mutants. Besides, surface LapA content was undetectable in Δ fleQ Δ lapE+control and Δ fleQ Δ lapE+wspR, and their total LapA was significantly lower than that in wild type, but similar to that in Δ fleQ+control. These results demonstrated that FleQ was essential for the c-di-GMP-mediated biofilm formation and LapA secretion. These results have been added to the new manuscript. The revision is shown in pages 21-22 lines 426-441 in the marked-up manuscript.

32. Figure 7C shows that cdG levels do not affect cAMP levels in a Δ fleQ strain and Δ fleQ + WspR strain. However, you cannot necessarily say that this is due to a loss in CyaA function. To support this claim, you would need to add Δ *cyaA* and Δ *cyaA* + WspR strains. Alternatively, you could also do a qRT-PCR analysis with the original strains from Figure 7C to demonstrate that *cyaA* expression is decreased (or not) in the Δ fleQ and Δ fleQ + WspR strains.

Our response:

Two methods were used to test the influence of fleQ on expression of *cyaA*, qRT-PCR analysis (Fig. 2) and promoter activity assay (Fig. 5 and Fig. 6), and both

results showed that expression of *cyaA* in Δ fleQ was higher than that in WT. *CyaA* promoter activity assay in Δ fleQ+control and Δ fleQ+wspR had also been done, and the results were shown in Fig. 5. *CyaA* promoter activity in Δ fleQ+control showed no obvious influence from that in Δ fleQ+wspR, indicating that c-di-GMP fails to regulate expression of *cyaA* in Δ fleQ. Consist with the expressional result, the cAMP level in Δ fleQ was higher than that in WT. High c-di-GMP level in WT caused lower cAMP, but c-di-GMP failed to affect cAMP level in a Δ fleQ strain, thus, our conclusion is that the c-di-GMP-mediated lowering of cAMP content is FleQ dependent.

33. Pls check the writing in the Discussion, which is a bit wordy and long.

Our response:

We have carefully revised the discussion part, and deleted some unnecessary descriptions. The discussion part has been shortened from 1565 words to 1208 words. The revision is shown in pages 23-30 lines 464-613 in the marked-up manuscript.

Response to reviewer 2:

Reviewer #2 (Comments for the Author):

The revised manuscripts describes genes that are regulated by the bacterial second messenger c-di-GMP and the transcription factor FleQ in *Pseudomonas putida*. Overall, the revisions address some of the major points that were raised during the initial review. One main caveat, the lack of biological replicates in the initially transcriptomic analysis, is addressed in the text and validation experiments on selected targets indicate the robustness of those specific results. The results refine what is known of the cellular programs that are controlled by c-di-GMP in this organism.

Our response:

Thanks a lot for all the comments. We do realize that using technical replicates for RNAseq is not reliable, and this is a weak experimental design, and it will not happen again in our future study. Function and regulation of several target genes from the RNA-seq results are presently under deeper research, and qRT-PCR is done first to confirm that these genes are regulated by c-di-GMP/FleQ.

Specific comments:

1. Line 131: "...c-di-GMP is involved..." ('is' is missing).

Our response:

The "...that c-di-GMP involved in." has been revised as "...that c-di-GMP is involved in" in the new manuscript. The revision is shown in page 8 lines 147 and 150 in the marked-up manuscript.

2. Lines 150-153: Please check this sentence for content and clarity.

Our response:

The sentence has been revised as "The first transcriptomic analysis (WT+wspR vs WT+control) above identified the potential genes regulated by c-di-GMP, and the second transcriptomic analysis (Δ fleQ vs WT) identified the potential genes regulated by FleQ. Thus, the genes co-regulated by c-di-GMP and FleQ should be found in both the first and the second transcriptomic analysis. The revision is shown in page 9 lines 175-181 in the marked-up manuscript.

3. Line 232 and 233: 'was' should be 'were' when referring to the c-di-GMP levels.

Our response:

The "was" has been changed to "were" when referring to the c-di-GMP levels in the new manuscript. The revision is shown in page 14 lines 276-277 in the marked-up manuscript.

4. Line 302: 'investigate' should be 'investigation'.

Our response:

The sentence has been revised as “BLAST results revealed that PP_0681, PP_0788, and PP_5586 encoded putative function-unknown proteins, but lapE and cyaA did not, thus lapE and cyaA were further investigated.” in the new manuscript. The revision is shown in page 19 lines 378-383 in the marked-up manuscript.

5. Lines 382/383: '...genes have not been identified...'

Our response:

The “...genes were not been identified...” has been changed to “...genes have not been identified...” in the new manuscript. The revision is shown in page 24 line 492 in the marked-up manuscript.

6. Line 384: '...in this study have not been identified...'

Our response:

The “...in this study were not been identified...” has been replaced by “...in this study have not been identified...” in the new manuscript. The revision is shown in page 24 line 494 in the marked-up manuscript.

7. Line 385: '...may be caused by the different...'

Our response:

The sentence has been revised as “This results discrepancy may be caused by the different type of the applied methods and procedures of transcriptomic analysis vs ChIP-seq analysis.experiment applied.” in the new manuscript. The revision is shown in page 24 line 494-496 in the marked-up manuscript.

8. Line 391: 'finds' should be 'finding'.

Our response:

The “finds” has been changed to “finding” in the new manuscript. The revision is shown in page 25 line 502 in the marked-up manuscript.

9. Line 456: That LapE is required and limiting in LapA secretion has been described previously in *P. fluorescens* (DOI: 10.1128/JB.00734-17). While the authors focus here on the multiple regulatory levels of adhesin function involving c-di-GMP, the previous work provides some mechanistic context. The authors cited this paper previously in the text, so this is just a second reference point, connecting their work with a previous finding.

Our response:

Thanks a lot for this comment. The study in *P. fluorescens* has been cited in the results part to introduce the reported function of *lapE*, thus it is unnecessary to repeat the context in discussion part. We have deleted those repeated description in discussion part, and focused on the multiple regulatory levels of adhesion function involving c-di-GMP in the new manuscript. The revision is shown in page 27 lines 552-561 in the marked-up manuscript.

10.Line 485: 'rest' should be 'remaining'.

Our response:

The “rest” has been replaced by “remaining” in the new manuscript. The revision is shown in page 29 line 597 in the marked-up manuscript.

April 10, 2021

Prof. Wen-Li Chen
Huazhong Agricultural University
State Key Laboratory of Agricultural Microbiology
Shizishan No.1
Wuhan, 44 430070
China

Re: mSystems00295-21 (Identification of c-di-GMP/FleQ-regulated new target genes including *cyaA* encoding adenylate cyclase in *Pseudomonas putida*)

Dear Prof. Wen-Li Chen:

I have reviewed the responses to the reviewers and the revised manuscript and believe all of the reviewers comments have been addressed either in the response or in edits to the manuscript. Therefore I am recommending acceptance.

Your manuscript has been accepted, and I am forwarding it to the ASM Journals Department for publication. For your reference, ASM Journals' address is given below. Before it can be scheduled for publication, your manuscript will be checked by the mSystems senior production editor, Ellie Ghatineh, to make sure that all elements meet the technical requirements for publication. She will contact you if anything needs to be revised before copyediting and production can begin. Otherwise, you will be notified when your proofs are ready to be viewed.

- Minimum resolution of 1280 x 720
- .mov or .mp4. video format
- Provide video in the highest quality possible, but do not exceed 1080p
- Provide a still/profile picture that is 640 (w) x 720 (h) max

We recognize that the video files can become quite large, and so to avoid quality loss ASM suggests sending the video file via <https://www.wetransfer.com/>. When you have a final version of the video and the still ready to share, please send it to Ellie Ghatineh at eghatineh@asmusa.org.

Sincerely,

Jonathan Eisen
Editor, mSystems

Journals Department
Table S4: Accept
Table S2: Accept
Table S5: Accept
Fig. S2: Accept
Table S1: Accept
Fig. S1: Accept
Table S3: Accept
Fig. S3: Accept
Table S6: Accept